# Simultaneous observations of NLC and MSE at midlatitudes: Implications for formation and advection of ice particles

Michael Gerding[1], Jochen Zöllner[1,*], Marius Zecha[1], Kathrin Baumgarten[1], Josef Höffner[1], Gunter Stober[1], and Franz-Josef Lübken[1]

[1]Leibniz-Institute of Atmospheric Physics at Rostock University, Kühlungsborn, Germany
[*]now at Planet AI GmbH, Rostock, Germany

*Correspondence to:* Michael Gerding (gerding@iap-kborn.de)

**Abstract.** We combined ground-based lidar observations of Noctilucent Clouds (NLC) with collocated, simultaneous radar observations of Mesospheric Summer Echoes (MSE) in order to compare ice cloud altitudes at a mid-latitude site (Kühlungsborn/Germany, 54° N, 12° E). Lidar observations are limited to larger particles ($>10$ nm), while radars are also sensitive to small particles ($<10$ nm), but require sufficient ionization and turbulence at the ice cloud altitudes. The combined lidar and radar data set thus includes some information on the size distribution within the cloud and by this on the 'history' of the cloud. The soundings for this study are carried out by the IAP RMR lidar and the OSWIN VHF radar. On average, there is no difference between the lower edges ($z_{\mathrm{NLC}}^{\mathrm{low}}$ and $z_{\mathrm{MSE}}^{\mathrm{low}}$). The mean difference of the upper edges $z_{\mathrm{NLC}}^{\mathrm{up}}$ and $z_{\mathrm{MSE}}^{\mathrm{up}}$ is $\sim$500 m, which is much less than expected from observations at higher latitudes. In contrast to high latitudes, the MSE above our location typically do not reach much higher than the NLC. In addition to earlier studies from our site, this gives additional evidence for the supposition that clouds containing large enough particles to be observed by lidar are not formed locally but are advected from higher latitudes. During the advection process, the smaller particles in the upper part of the cloud either grow and sediment, or they sublimate. Both processes result in a thinning of the layer. High-altitude MSE, usually indicating nucleation of ice particles, are rarely observed in conjunction with lidar observations of NLC at Kühlungsborn.

## 1 Introduction

Noctilucent Clouds (NLC, also known as Polar Mesospheric Clouds, PMC) and Polar Mesospheric Summer Echoes (PMSE) have been observed since several decades mainly in the polar regions by ground-based and space-based instruments (e.g. DeLand et al., 2003; Chu et al., 2003; Morris et al., 2007; Collins et al., 2009; Latteck and Bremer, 2017). Even older data sets of NLC exist from visual observations (e.g. Leslie, 1885). Observations at mid latitudes showed that mesospheric ice clouds can exist also equatorward of 60° latitude and occasionally even equatorward of 45° latitude (Thomas et al., 1994; Chilson et al., 1997; Wickwar et al., 2002; Ogawa et al., 2011; Russell et al., 2014). First simultaneous soundings of both phenomena have been achieved by Nussbaumer et al. (1996) at the ALOMAR observatory at 69° N. The observations stimulated the impression that both phenomena are related to ice clouds in the mesopause region, even if there are some differences in occurrence and vertical extension. NLC are usually observed between $\sim$80-86 km (e.g. Fiedler et al., 2017), while PMSE stretch higher

and appear at ∼80-90 km (e.g. Latteck and Bremer, 2017). Further studies revealed that PMSE additionally require sufficient ionization of the ambient air to get the ice particles charged. Additionally, scattering of the radar wave occurs only on structures in the plasma that are produced by turbulence but can persist even if the turbulence ceased. Rapp and Lübken (2004) published a review of PMSE physics and their relation to ice clouds in the mesopause region. Thus, NLC and PMSE are both indicators for
temperatures below the frost point, i.e. the ice clouds provide indirect information on temperature in a region of the atmosphere where other data is sparse.

In this study we utilize combined observations by lidar and radar to gather information about the origin of the NLC layer at midlatitudes. There is some debate about the role of advection from higher latitudes compared to local ice particle formation. This is in particular important for the interpretation of trends in midlatitude ice clouds (e.g. Thomas, 2003; Russell et al., 2014).
Some studies found a strong dependence of ice observations on equatorward directed wind (Morris et al., 2007; Zeller et al., 2009; Gerding et al., 2013b) or planetary wave activity (Nielsen et al., 2011), while other studies explained the observations mainly by local temperature structure (e.g. Herron et al., 2007; Hultgren et al., 2011; Stevens et al., 2017). Simultaneous observations by lidar and radar give additional information on this topic due to their different size dependencies. Lidars are mainly sensitive to ice particles with diameters of some ten nanometers (NLC), while radar echoes ((P)MSE), ionization and
turbulence provided, indicate small or large ice particles, where smaller particles may be freshly formed in the mesopause region and just start to sediment. Local NLC formation therefore implicates the simultaneous existence of freshly formed particles, i.e. of typically an MSE layer extending above the NLC. If the advection of NLC dominates, the initial ice particles should have already sedimented and grown.

Simultaneous observations of NLC and (P)MSE to solve this question are technically challenging and rare. They require a
powerful lidar and a VHF radar being co-located. The lidar needs to be daylight-capable because (P)MSE are mainly limited to daylight conditions (e.g. Thomas et al., 1996; Zecha et al., 2003; Rapp and Lübken, 2004). During darkness and outside the auroral oval, the ionization in the D-region is typically too small to create radar echoes in the mesopause region. Additionally, the lidar needs to be sensitive enough for the typically weak NLC backscatter signals. So far, the only statistical studies on joint NLC and (P)MSE occurrence and layer parameters from simultaneous observations by lidar and radar are performed at polar
latitudes (e.g. von Zahn and Bremer, 1999; Klekociuk et al., 2008; Kaifler et al., 2011). For most of the time of NLC detection also PMSE have been observed, while, on the contrary many PMSE have been detected in the absence of NLC. Typically, PMSE and NLC layers have very similar lower edges, but PMSE stretch several kilometers higher than NLC. Li et al. (2010) found in PMSE observations that average ice particle radii are smaller above 85 km than below. This can be explained by local ice formation in the high latitude mesopause region (observed as PMSE), happening in parallel to the occurrence of larger ice
particles below 85 km (observed as NLC and PMSE). These main layer properties are similar in northern and southern polar regions, even though observations at Davis (69° S) show typically less and weaker echoes compared to ALOMAR (69° N) (Morris et al., 2007; Latteck and Bremer, 2017). For midlatitudes, either only MSE or NLC statistics have been described so far. Midlatitude MSE layer properties have been published by Latteck et al. (1999) and Zecha et al. (2003) based on data of the OSWIN radar at Kühlungsborn (Germany, 54°N). They found a much lower occurrence rate compared to high latitudes in
the Northern Hemisphere, but a similar altitude distribution. Midlatitude NLC layer properties have been described by Gerding

et al. (2013a), using the Rayleigh-Mie-Raman lidar co-located with OSWIN. Similarly, they found a much smaller occurrence rate compared to high latitudes, but comparable NLC altitudes. Nevertheless, a joint examination of lidar and radar observations at our site is lacking.

In this paper we compare NLC and MSE layer properties such as lower and upper edges for all periods with simultaneous observations. By this we concentrate on events when at least part of the ice particles have grown to sizes of some ten nanometers. By the combination of NLC and MSE signals we also avoid confusion of the ice-related MSE with other mesospheric echoes in summer that are sometimes observed at much lower latitudes, i.e. at much too high temperatures for ice existence (Muraoka et al., 1989; Kubo et al., 1997).

In the following Section 2 we describe the instrumentation and the available data set. The vertical distributions of layer edges and maxima are presented in Section 3. In Section 4 we examine the influence of local wind and temperature profiles on the ice NLC/MSE. In Section 5 we compare our results to data from polar latitudes, leading to our conclusion about the relevance of advection for NLC occurrence at our site.

## 2    Methodology and single layer comparison

The NLC and MSE observations used here are made at Kühlungsborn (54° N, 12° E) with the daylight-capable Rayleigh-Mie-Raman (RMR) lidar and the OSWIN VHF radar, respectively. Additional data are provided by the co-located potassium resonance lidar measuring temperatures above the NLC (e.g. von Zahn and Höffner, 1996; Alpers et al., 2004) and by the nearby (120 km to the north-east) meteor radar at Juliusruh (55° N, 13° E) (Stober et al., 2012, 2017). In this section we describe the main instruments, show some examples for edge detection and the general appearance of NLC and MSE, and give an overview on the used data and its representativeness.

### 2.1    NLC observations by the daylight-capable IAP RMR lidar at Kühlungsborn

The daylight-capable RMR lidar started operation in summer 2010 and replaced the former RMR lidar used for nighttime NLC observations. A general description of the lidar has been given by Gerding et al. (2016). In summary the lidar uses a frequency-doubled Nd:YAG laser at 532 nm with ∼20 W average power. The backscatter signal is collected by a single telescope of 80 cm diameter and detected by an Avalanche Photo Diode. The daylight capability is achieved by a field of view of the telescope of only ∼60 μrad, a narrow-band interference filter (130 pm), and a double Fabry-Pérot etalon (∼4 pm full spectral width at half maximum). The lidar is designed for observation of middle atmosphere temperatures and their variability due to gravity waves and tides (Kopp et al., 2015; Baumgarten et al., 2018) and for detection of NLC in summer. For NLC measurements, aerosol and molecular scattering are separated by exponential interpolation of the background-corrected Rayleigh (molecular) backscatter signal above and below the cloud. The NLC backscatter coefficient at 532 nm ($\beta_{532}$) is then calculated from the aerosol backscatter normalized to the molecular backscatter, the molecular backscatter cross section, and a reference air density to quantify the cloud brightness. Under typical transmission conditions and during full daylight the sensitivity limit of the lidar for NLC is at $\beta_{532} = 0.3 \cdot 10^{-10}\,\mathrm{m^{-1}sr^{-1}}$ (in the following we describe $\beta_{532}$ in units of $10^{-10}\,\mathrm{m^{-1}sr^{-1}}$, i.e. here $\beta = 0.3$).

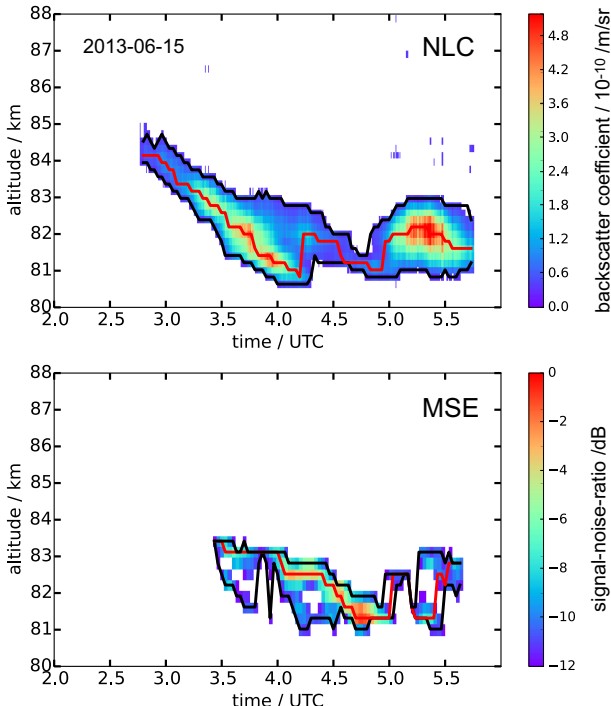

**Figure 1.** Examples for detection of NLC (top) and MSE (bottom) using observations on 15 June 2013. The black lines show the upper and lower edges ($z^{\mathrm{low}}$ and $z^{\mathrm{up}}$), the red line the layer maxima ($z^{\mathrm{max}}$). Edge detection is done on the temporal resolution of the radar (2 min). Edges are defined at $\beta = 0.3$ (NLC) and SNR = -12 dB (MSE).

For this study, the backscatter signal has been integrated for 30 s and smoothed by a running average over 15 min. The vertical resolution is set to 195 m. The individual profiles have been manually inspected for NLC and only positively identified profiles are used for further processing (cf. Gerding et al., 2013a). NLC backscatter maxima ($z_{\mathrm{NLC}}^{\mathrm{max}}$) and layer edges ($z_{\mathrm{NLC}}^{\mathrm{low}}$ and $z_{\mathrm{NLC}}^{\mathrm{up}}$) are evaluated automatically. A typical NLC case is shown in Figure 1 (upper panel). The layer edges, defined at $\beta = 0.3$, and layer maxima are identified by an algorithm and marked in the Figure by black and red lines.       **Fig.1**

## 2.2 MSE observations by the OSWIN VHF radar at Kühlungsborn

The monostatic OSWIN VHF radar (53.5 MHz) operated in an unattended and continuous measurement mode during the summer seasons. Until 2013 a phased-array antenna field consisting of 12 x 12 Yagi antennas was used. The beam could be tilted, but for the comparisons of MSE and NLC only the vertically directed beam with a beam width of $6°$ was selected. Two 16-bit complementary codes with $2\,\mu s$ pulse elements were used. The repetition frequency was set to 1200 Hz. For reception, the antenna array was split in six subgroups with 24 antennas each, which were connected to six receivers. Data points were created by coherent integrations of 20 samples. Time series of 1024 data points are acquired within 34.1 s. Considering the time of further alternating measurements, the time resolution for MSE observations is 2 min. After 2013 the antenna array was

refurbished. The new array is based on 133 Yagi antennas arranged in a hexagonal structure. The width of the vertically directed beam is about 6° again. Two 32-bit-complementary codes with 2 $\mu$s pulse elements and 625 Hz repetition frequency are used. For reception, six subgroups of 21 antennas each are connected to six receivers. Time series of 1024 samples (inclusive of eight coherent integrations) result in length of 26.2 s. In summary we assume fairly similar technical conditions regarding the radar measurements during the summer seasons. In both periods a height resolution of 300 m is maintained. The backscattered signals received by the six receivers are combined phase conform. Signal-to-noise ratios (SNR) are estimated from the autocorrelation functions of the time series in each height channel. As we do not have an absolute calibration of the radar, we use SNR as an approximation for the echo intensity. The lowest signal level used for MSE observations is chosen at an SNR of -12 dB. A typical example with identified edges is presented in Fig. 1 (lower panel).

## 2.3 Examples for simultaneous NLC-MSE observations

In the following we show different cases of simultaneous NLC-MSE observations to demonstrate the variability of the layers. Similar to previous studies we often find very good agreement between NLC and MSE, while there are differences in other cases (cf. von Zahn and Bremer, 1999; Klekociuk et al., 2008; Kaifler et al., 2011). Examples are given in Figure 2 . Figure 2 a) **Fig.2** shows an event that was observed on 17 June 2010. While the NLC (filled contours) was first detected above the limit of $\beta = 0.3$ at 2:45 UTC, the MSE (contour lines) was only observed after 3:30 UTC, when the solar elevation exceeded $\sim$5° and the ionization of the atmosphere was large enough to allow a radar backscatter signal. Then both phenomena follow the same vertical movement, presumably related to the cold phase of a gravity wave, with the MSE sometimes reaching to higher altitudes.

Also on 10 July 2015 (Fig. 2 b) NLC and MSE showed a good agreement. The lidar was switched on at 13:50 UTC when the MSE already existed, and NLC were observed from the beginning of the lidar sounding, but in a smaller altitude range. Between 15:30 and 16:50 UTC the NLC vanished, but the MSE showed only a short gap of 10 min and set in again at a much higher altitude about half an hour before the NLC occurs again. Then both layers agreed well until the MSE disappeared at $\sim$19 UTC at a solar elevation of $\sim$4°. The NLC observation continued for another 5 h, until the layer descended below 80 km at 0 UTC.

On 10 June 2011 (Fig. 2 c) NLC were observed from the beginning of the lidar sounding at 5:21 UTC. Earlier, the ice particles were already detected by the radar as MSE, but tropospheric clouds inhibited lidar operation. In contrast to the other cases, here the MSE extended above the NLC and was in fact strongest at altitudes above the NLC upper edge. Later on, the NLC ceased while the MSE continued.

A rare example of a double ice layer was observed on 11 June 2012 (Fig. 2 d). The cloud was first confirmed by the lidar, showing that the NLC was very variable in altitude due to the presence of gravity waves. After $\sim$3:20 UTC the NLC layer thickness increased to $\sim$5 km (81 – 86 km). Just at the same time the radar echo started due to increasing solar elevation, i.e. increasing ionization. The MSE quickly grew into a double layer, with a gap around 83 km altitude, where the NLC was found brightest. The NLC and MSE around 82 km faded away at 4:23 UTC and set in again at 4:40/4:45 UTC (MSE/NLC). In the meantime, the only remaining ice signal was observed by the radar above 83 km. Past $\sim$5 UTC the upper MSE layer vanished,

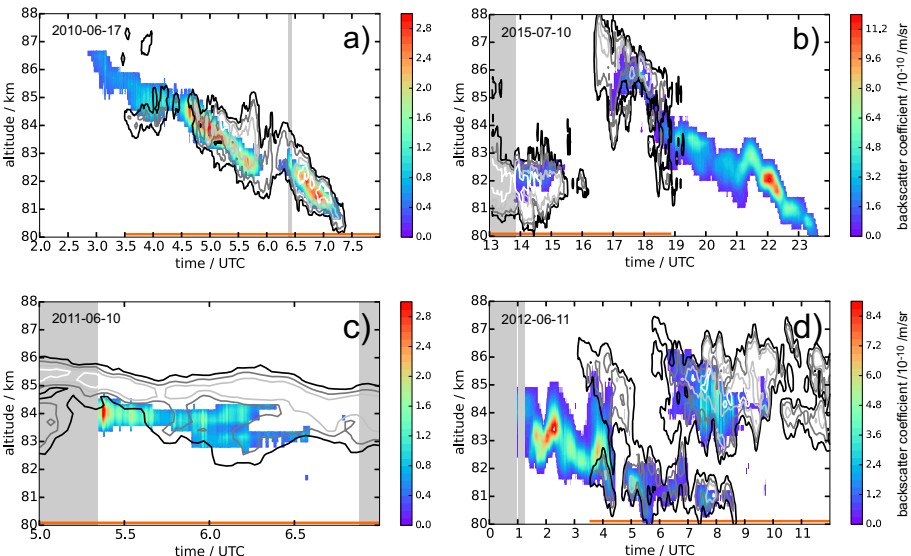

**Figure 2.** Examples for observations of MSE (contour lines at -12/-6/0/6 dB, if applicable) and NLC (filled contours with scale on the right). The orange lines indicate periods with solar elevation $>5°$. The gray shaded areas mark periods without lidar soundings due to presence of clouds. a) 17 June 2010: MSE starts after sunrise into existing NLC. b) 10 July 2015: MSE vanishes after sunset while NLC continuous. c) 10 June 2011: When lidar starts operation NLC is immediately detected, but does not extend as high as the MSE. d) 11 June 2012: MSE starts into existing NLC after sunrise and continues especially at higher altitudes when NLC disappears.

but reappeared shortly after. At $\sim$6:15 UTC also the NLC formed a (very rare) double layer until $\sim$8:30 UTC, when the lower layer of both, NLC and MSE, ceased. Despite being weak, the upper NLC layer remained observable until $\sim$10 UTC, while the MSE still continued in a broad range (83 – 87 km). The variable structure of the ice cloud with double layers indicates a highly dynamic behavior of the atmosphere with presence of strong gravity waves. Nevertheless, a detailed examination of the

5   dynamical structure is beyond the scope of this paper.

The examples shown above demonstrate the different relations of the NLC and MSE layer edges and the different degrees of accordance of the layers. This is in general agreement with observations at polar latitudes (e.g. Klekociuk et al., 2008; Kaifler et al., 2011). The examples indicate an often good concurrence of the lower edges but a worse agreement of the upper edges. If solar elevation (i.e. ionization) is sufficiently large, NLC are often but not always accompanied by MSE. The latter might be

10   explained by the radar detection threshold or missing turbulence, but this cannot be checked here because a lack of appropriate measurements. Periods with MSE but absent NLC can be caused by mainly small ice particles, resulting in lidar signals below the NLC detection threshold. In the following we neglect profiles of NLC without MSE as well as MSE without NLC to be sure that for this study all requirements for the observation of small and large ice particles are fulfilled (see below).

## 2.4 The data set used for this study

Within this study we only focus on simultaneous NLC and MSE events (e.g., after 3:30 UTC in Fig. 2 a, but excluding the little MSE gaps between $4-5$ UTC), to compare the altitude ranges where both phenomena are observed. Nighttime NLC, where ionization of the atmosphere is typically too small for MSE generation, are ignored as well as a few NLC where the radar was
switched off for maintenance. Vice versa we do not count ice clouds (detected as MSE) that are found too weak to be observed by lidar or that occured when the lidar was switched off, e.g., during tropospheric cloud coverage. Overall, we use $\sim$67 h of NLC with $\beta > 0.3$ for this study, out of 188.5 h of NLC in total ($\beta > 0$, day and night) in the years $2010-2016$. About 121 h of NLC detection cannot be used here because of either too weak NLC ($\beta < 0.3$, $\sim$20%) or the absence of MSE. MSE get sparse at low solar elevation, because of missing ionization ($\sim$35% of the time the solar elevation is below 5°). Furthermore, either
turbulence can be missing or the radar detection threshold can be too high. Nevertheless, this subset of NLC is representative for the whole data set in terms of layer heights, as discussed in Section 5. The total usable lidar operation time within the seven summers (1 June to 4 August) is 3337 h. MSE are observed by OSWIN for 960 h out of 8600 h of total sounding time. As mentioned above, only for 67 h the MSE (SNR > 12 dB) occur during lidar operation and simultaneously with NLC of $\beta > 0.3$. These data are distributed across 31 days with an average ice cloud duration of 2.2 h. For this study it is not relevant whether
the ice observation is uninterrupted in time or not, because the layer parameters are derived based on individual (but smoothed) profiles. Note that this study is representative for ice clouds containing sufficiently large particles to be detectable by lidar, but it is not representative for MSE (ice clouds) in general.

## 3 Comparison of NLC and MSE layer properties

Based on the data set of simultaneous NLC and MSE (gridded to 2 min temporal resolution) we identify the lower edges ($z_{\mathrm{NLC}}^{\mathrm{low}}$
and $z_{\mathrm{MSE}}^{\mathrm{low}}$), the altitudes of maximum brightness ($z_{\mathrm{NLC}}^{\mathrm{max}}$ and $z_{\mathrm{MSE}}^{\mathrm{max}}$) and the upper edges ($z_{\mathrm{NLC}}^{\mathrm{up}}$ and $z_{\mathrm{MSE}}^{\mathrm{up}}$) for each profile of both phenomena independently, as shown in Fig. 1. In the rare case of a double layer we take the lower edge of the lower layer and the upper edge of the upper layer together with the absolute maximum. Overall, we get 1931 profiles with their respective properties, even if the particular smoothing of lidar and radar data needs to be taken into account for interpretation. In Figure 3 (left) the altitude distributions of $z_{\mathrm{NLC}}^{\mathrm{low}}$ and $z_{\mathrm{MSE}}^{\mathrm{low}}$ are summarized in 1 km bins. The most striking feature is the   **Fig.3**
similarity between the altitude distributions of NLC and MSE lower edges. In the evaluated data set, no cloud is observed below 80 km. The mean values of both, $z_{\mathrm{NLC}}^{\mathrm{low}}$ and $z_{\mathrm{MSE}}^{\mathrm{low}}$ are found at $\sim$82.6 km (dashed lines). The figure shows only small differences between both distributions. In Figure 3 (right) the lower edges of all individual NLC/MSE profiles are compared. The plot confirms the similarity between both parameters. Most often the $z^{\mathrm{low}}$ agree within a single altitude bin, which also shows up in the correlation coefficient of R=0.79. The differences between $z_{\mathrm{NLC}}^{\mathrm{low}}$ and $z_{\mathrm{MSE}}^{\mathrm{low}}$ can be up to a few hundred meters,
and there is no altitude dependence of the differences. Thus, very high ice clouds show the same similarity as low or typical layers. The mean difference between $z_{\mathrm{NLC}}^{\mathrm{low}}$ and $z_{\mathrm{MSE}}^{\mathrm{low}}$ is $\sim$40 m with a standard deviation of 417 m. In a few cases the altitude difference can be up to $4-5$ km. This can already be seen in Fig. 2 d), e.g., in cases of MSE onset in the morning twilight where sometimes the MSE only agrees with the uppermost part (i.e. largest ionization) of the ice cloud. Part of the differences

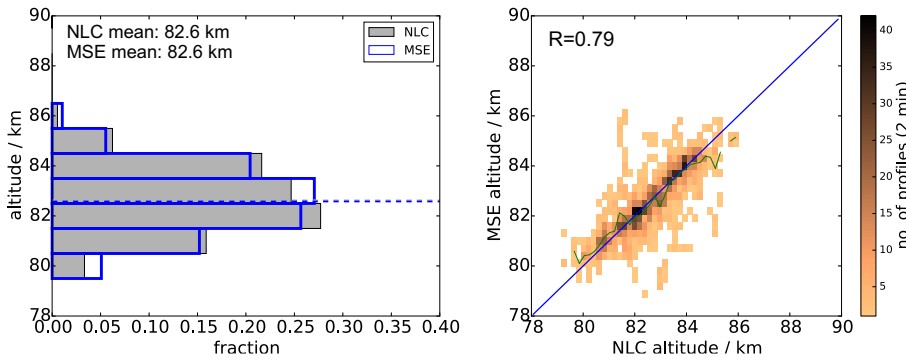

**Figure 3.** Comparison of lower edges of NLC and MSE ($z_{\text{NLC}}^{\text{low}}$ and $z_{\text{MSE}}^{\text{low}}$). a) Histogram with NLC edges in grey (filled) and MSE edges in blue (open histogram). Mean values are indicated by the dashed lines. b) Scatter plot of MSE and NLC edges. Only simultaneous soundings based on a 2 minute temporal resolution are evaluated. The green line in b) shows the average MSE lower edge altitude for each NLC edge bin and the blue line indicates identical altitudes.

can also be explained by the different fields of view (FOV). The radar has a FOV of $6° = 0.1$ rad, while the lidar FOV is only ~1/1700 of this (62 $\mu$rad). This may lead to some differences at least in cases of very structured ice clouds, because any feature in the ice cloud needs some time for drifting through the radar FOV, e.g. ~3.5 min at 20 m/s wind speed. Rarely, the different size dependencies of lidar and radar signals can lead to MSE even a few km below the NLC. On average, these effects only 5    have a small influence on the general distribution.

The altitude distribution of the ice layer maxima (backscatter coefficient for NLC and signal-to-noise ratio for MSE) is shown in Figure 4 . The mean of $z_{\text{NLC}}^{\text{max}}$ is observed at 83.3 km (grey dashed line in left panel), while the mean of $z_{\text{MSE}}^{\text{max}}$ is slightly    **Fig.4** higher at 83.6 km (blue dashed line in left panel). No NLC maximum is observed above 86 km, but a few MSE maxima. This is also resembled in the scatter plot (Fig. 4, right). Similar to the lower edges there is no pronounced altitude dependence of 10   the differences between $z_{\text{NLC}}^{\text{max}}$ and $z_{\text{MSE}}^{\text{max}}$ (green line). The correlation coefficient is R=0.80. While again the individual $z_{\text{NLC}}^{\text{max}}$ and $z_{\text{MSE}}^{\text{max}}$ sometimes differ by a few kilometers, the mean value of the differences is only ~300 m with a standard deviation of 375 m.

The largest differences between NLC and MSE are expected at the upper edges of the layer (cf. Kaifler et al., 2011). This is due to the fact that the nucleation typically starts close to the mesopause, where water vapour saturation is largest. These 15   small particles can already be detected by radars (signal strength proportional $r^2$, if turbulence and ionization allow). If the supersaturated region extends further down, the particles start to sediment and grow, and become finally visible for lidars (signal strength proportional $r^5..r^6$). Indeed, we find the mean $z_{\text{MSE}}^{\text{up}}$ about 500 m above the mean $z_{\text{NLC}}^{\text{up}}$ for all simultaneous observations (Figure 5 ), i.e. at 84.5 km and 84.0 km, respectively. The scatter plot shows that the height difference is largely    **Fig.5** independent of altitude. Only for the very high (>85 km) and very low (<82 km) layers the differences seem to vanish, but the 20   number of such events is small. The smaller height difference can be explained by the typically smaller width of the ice clouds at these altitudes (not shown). In contrast to the layer maxima and lower edges, differences of a few kilometers are mainly

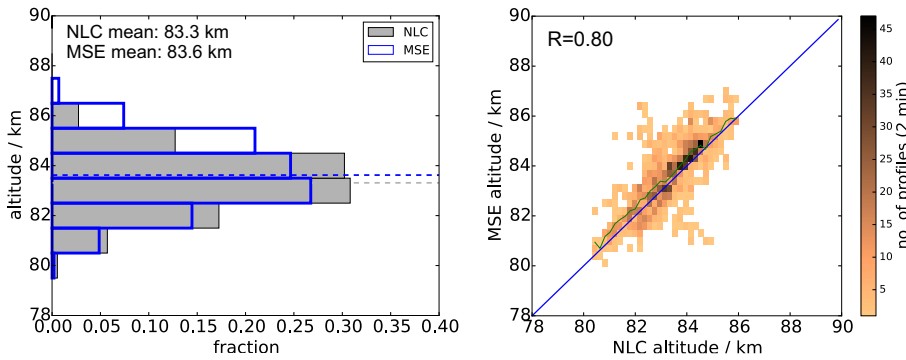

**Figure 4.** Same as Figure 3, but for the layer maxima ($z_{\mathrm{NLC}}^{\max}$ and $z_{\mathrm{MSE}}^{\max}$).

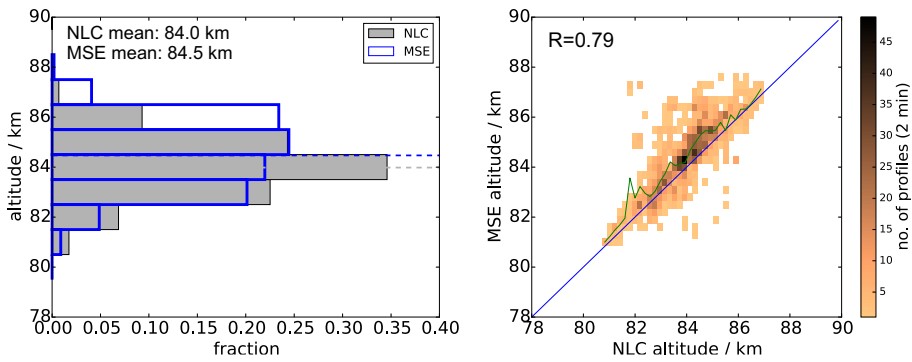

**Figure 5.** Same as Figure 3, but for the upper edges of NLC and MSE ($z_{\mathrm{NLC}}^{\mathrm{up}}$ and $z_{\mathrm{MSE}}^{\mathrm{up}}$).

found with the MSE top being much higher than the NLC top. Thus, the distribution of differences is not symmetric, but has a tail towards higher MSE as typically expected. Note that the correlation of $z_{\mathrm{MSE}}^{\mathrm{up}}$ and $z_{\mathrm{NLC}}^{\mathrm{up}}$ is still high (R=0.79).

From the upper and lower edges the NLC and MSE layer thicknesses can easily be calculated. The results are shown in the histograms in Figure 6 . Typically, the NLC thickness ($z_{\mathrm{NLC}}^{\mathrm{up}} - z_{\mathrm{NLC}}^{\mathrm{low}}$) is below 2 km, with a mean value of 1.35 km. The MSE **Fig.6**
5    are typically slightly thicker, having a mean layer thickness of 1.89 km. These numbers are already expected from the difference of mean upper and lower edges. Furthermore, the histogram shows a larger quantity of thick MSE, having thicknesses of more than 4 km. That means that in a few cases the MSE width is much larger than the NLC width, but the average layer thicknesses differ by only $\sim$500 m.

## 4   Comparison of NLC/MSE with local wind and temperature structure

10   There is general agreement that ice clouds are limited to regions with temperatures below the frost point temperature. For our location we have already shown that NLC occur only in the cold phases of gravity and planetary waves, while mean temperatures are above the frost point in the whole mesopause region (Gerding et al., 2007, 2013a). Additionally, we demonstrated that

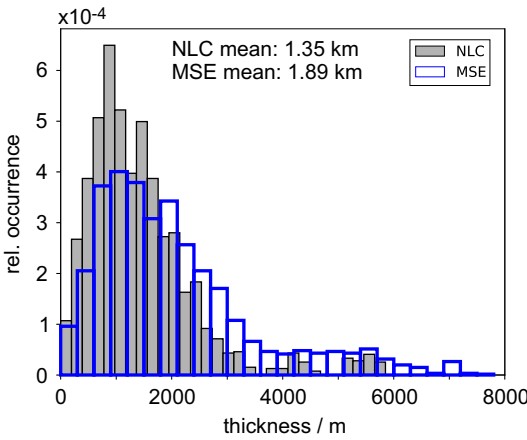

**Figure 6.** Histogram of layer thicknesses, calculated as $z_{\text{NLC}}^{\text{up}} - z_{\text{NLC}}^{\text{low}}$ for NLC (grey, filled) and $z_{\text{MSE}}^{\text{up}} - z_{\text{MSE}}^{\text{low}}$ for MSE (blue, open). The relative occurrence (O) is normalized such that all bins times the bin width ($\Delta$) sum to 1, i.e. $\sum (O \cdot \Delta) = 1$. Bin widths are 195 m for NLC and 300 m for MSE.

southward directed winds are necessary for the occurrence of NLC at our site (Gerding et al., 2007, 2013b). Here we want to check whether the ambient wind and temperature conditions are responsible for the altitude of the layer edges, i.e. the thickness of the NLC and MSE. We therefore evaluated the temperatures and winds especially above the layers.

Temperature soundings above MSE require a metal resonance lidar (for the altitude coverage) and daylight capabilities (for
observation during MSE). The potassium resonance lidar at Kühlungsborn was in operation until the end of 2012, i.e. it covered part of the examined time period. The upper panel of Figure 7 shows an example of the temperature structure above two ice   **Fig.7** layers in the early morning of 27 June 2011. The first layer ('Cloud 1') arose at ∼2:30 UTC in the lidar (NLC, coloured layer) and did not extend above 83.5 km. The MSE (grey contour lines) appeared first at 3:30 UTC in 84.3 km, when ionization was sufficient (solar elevation 4.7°). Above Cloud 1 the potassium resonance lidar observed temperatures of more than 150 K,
i.e. higher than the expected frost point temperature for these altitudes. In other words, the temperature structure inhibits an expansion of the layer to higher altitudes. Later, Cloud 1 descends and vanishes at 5:15 UTC. Another ice layer ('Cloud 2') appeared at ∼5:30 UTC around 86 km. This coincides with a strong temperature decrease below ∼140 K in the same altitude range. We point out that Cloud 2 was observed by radar only (MSE), but did not contain particles large enough to be observed by lidar (NLC). Lidar observations stopped due to tropospheric clouds at ∼8 UTC. Note the integration time of the temperature
data of 2 h, shortened to 1 h at the beginning and end of the sounding.

The lower panel of Fig. 7 shows the meridional wind observed by the nearby meteor radar (120 km north-east of Kühlungsborn). The warm region above Cloud 1 is accompanied by a northward wind, while the wind in the altitude of the layer is southward as expected from previous observations. At the time and altitude of the (higher) Cloud 2 the wind direction has changed to a weak southward wind. Overall, this typical case shows a large likelihood for advection of Cloud 1 that appeared
as both MSE and NLC, because local ice formation is inhibited by the high temperatures above 85 km. Note that before 02 UTC low temperatures of ∼140 K have been observed above 85 km, but at this time no ice cloud was observed. The second case

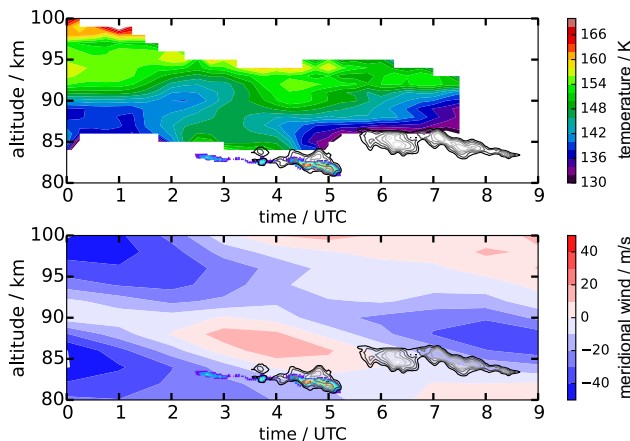

**Figure 7.** Temperature (top) and meridional wind (bottom) above Kühlungsborn during 26/27 June 2011. The MSE data is embedded in both panels for illustration as open grey contours and the NLC data in colored filled contours. Cloud 1: 2:30 – 5:15 UTC, Cloud 2: 5:30 – 8:40 UTC

(Cloud 2) has the potential for local formation above 85 km, but this layer is confined to higher altitudes and does not contain larger ice particles (NLC). Both clouds are confined to southward winds.

We have analysed the temperature and wind data set above the NLC/MSE for all available coincident measurements. Due to often hazy sky conditions and therefore large solar background at near-infrared wavelengths, the temperature data set of the potassium lidar is smaller than the NLC data set of the RMR lidar. Between 2010 and 2012 only seven events can be evaluated, with some of the data sets showing a gap of up to 2 km between the NLC/MSE and the temperature data. The temperature structure above the ice clouds varies between the events. Partly, we find an immediate temperature increase above (five cases), inhibiting ice existence in these heights. In the other two cases, the supersaturated region extends for some kilometers above the observed ice cloud and includes the height region expected for nucleation (87 – 90 km, e.g. Kiliani et al. (2013)). In this case, ice particles may still exist in the supersaturated region, even if not detected by the OSWIN radar. Low temperatures typically persist only for a few hours at our site (e.g. Gerding et al., 2007). In this time, ice particles only grow to a few nanometers radius (Rapp and Thomas, 2006), and continuous turbulence is needed to create radar echoes, in contrast to intermittent turbulence being sufficient in combination with larger ice particles (Rapp and Lübken, 2004). Additional ionization is needed in both cases. Unfortunately, there is no information about ambient turbulence available and the question about ice existence cannot be finally answered.

The wind structure above the NLC/MSE is also not uniform in all events. Overall, 23 events between 2010 and 2016 have been evaluated. During most of the events (nine cases) the wind above the ice layer is southward (as in the ice layer). Seven cases show northward winds above the cloud. In three cases the wind direction is changing, while four cases show only weak winds (less than ±10 m/s).

## 5 Discussion

Studies on the layer properties of NLC and MSE at midlatitudes are rare, because there is only a small occurrence frequency of $5-10\%$ for MSE and NLC (e.g. Thomas et al., 1996; Zecha et al., 2003; Gerding et al., 2013a). Therefore, analyses of average layer parameters need multi-year observations to yield a representative data base. The only statistical NLC study at midlatitudes has been done at our site at Kühlungsborn based on the nighttime observations of the previous RMR lidar (Gerding et al., 2013a). The results are in good agreement with the data presented here. They report a mean centroid height of 82.7 km which compares very well with the mean centroid height of 82.6 km and mean peak height of 82.8 km of all NLC (daytime and nighttime) in the 2010–2016 period used here. Selecting only days that also show MSE, the mean NLC peak height is slightly higher (83.0 km), but within the geophysical variability. The remaining difference to the mean $z_{\text{NLC}}^{\text{max}}$ of 83.3 km mentioned in Fig. 4 can be explained by the further selection of profiles really showing simultaneous MSE. Especially some very low NLC profiles (below 80 km) are excluded here because of missing MSE, which is potentially caused by the reduced electron density. Furthermore, the faintest NLC profiles with $\beta < 0.3$, e.g., in the beginning and end of events are also excluded in Fig. 4. Therefore the apparent difference in NLC layer heights compared to previous studies can be explained by geophysical variability and treatment of the NLC data. But this does not hamper the representativeness of this study for all NLC.

For MSE there is an earlier study by Zecha et al. (2003) using radar data from Kühlungsborn. They report the occurrence rate of MSE at each particular height bin which is different from the histogram of peak altitudes that we present here. Even more important, we limit the MSE data set to events with simultaneous NLC, because we want to focus on optically visible ice clouds. By this we suppress weak and high (typically >86 km) layers of MSE that are often not accompanied with NLC. We note that we skip the majority of MSE by this selection. Therefore, the results presented here are not representative for all MSE.

As explained above, there are only very few studies for high-latitude, simultaneous NLC and PMSE. Kaifler et al. (2011) evaluated a large data set of NLC and PMSE from ALOMAR (69° N, 16° E). Simultaneous events are summarized in their categories III, IV, and V. In their Table 3 they report also quasi-identical lower edges of NLC and PMSE, even if the $z^{\text{low}}$ at higher latitudes are observed 0.5 km below the midlatitude values. The identity of NLC and MSE lower edges is to some extend induced by the data selection criteria. Kaifler et al. (2011) used only NLC with a peak brightness of $\beta > 4$, shifting the lower edge a few hundred meters down compared to all NLC (their Fig. 5). We find a similar effect in our data set (not shown). This explains, together with the fact that NLC are thinner at midlatitudes, the altitude difference of NLC lower edges between high and midlatitudes.

Regarding the upper edges ($z_{\text{NLC}}^{\text{up}}$ and $z_{\text{PMSE}}^{\text{up}}$), Kaifler et al. (2011) report a mean difference of ~3.3 km, i.e. much larger than the difference of 0.5 km that we observe at midlatitudes. Klekociuk et al. (2008) examined one season of simultaneous observations of NLC and PMSE at 69° S at Davis Station (Antarctica). Occurrence rates of both phenomena are much smaller compared to the Northern Hemisphere. The authors do not provide numbers for the upper and lower edges of the layers, but from their Fig. 1 one can expect a difference of the upper edges similar to the numbers given by Kaifler et al. (2011). We therefore examined whether the small difference $z_{\text{MSE}}^{\text{up}} - z_{\text{NLC}}^{\text{up}}$ presented here is an arbitrary result of our limited data set. We

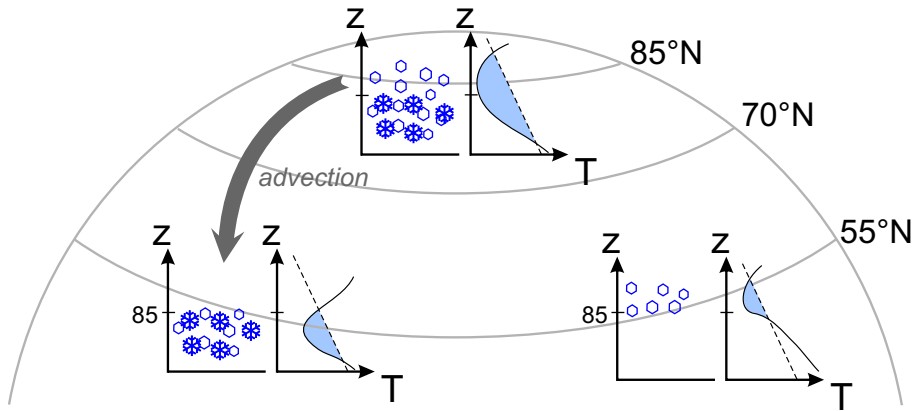

**Figure 8.** Schematic of the latitudinal differences for ice cloud formation. The x-y-plots represent the ice particle distribution with height (left) and the corresponding temperature profile (right). Small particles only visible for the radar are marked by light blue hexagons, larger particles by blue snowflakes. The light blue part of the temperature profile shows the region of supersaturation (dashed: frost-point temperature). The altitude of 85 km is marked, forming a typical upper limit of NLC.

chose different sub-data sets containing about half of the profiles, either by a random process or by selecting only the first or second half of the data. The small difference of upper edges persisted throughout all selection runs, varying between 0.2 km and 0.7 km.

The reason for the large height difference between $z_{\mathrm{NLC}}^{\mathrm{up}}$ and $z_{\mathrm{PMSE}}^{\mathrm{up}}$ at high latitudes is the typically much larger thickness
of the PMSE layer (cf. our Fig. 6 and Table 3 in Kaifler et al. (2011)). In agreement with these observations, Kiliani et al. (2013) demonstrated with a 3d trajectory model the formation of ice-particles at the high-latitude mesopause, and subsequent descent and growth. At high latitudes often a layer of small particles (only visible by radar) exists above the larger ice particles that can also be detected by lidar. This situation is in qualitative agreement with Odin/OSIRIS observations (Hultgren and Gumbel, 2014) and sketched in the upper part of Figure 8 . The small difference between upper layer edges in our observations **Fig.8**
suggests that the layer of small particles is missing at midlatitudes. In this case, the larger, optically visible ice particles cannot be formed locally, but typically have to be advected (see lower left part of Fig. 8). Kiliani et al. (2013) found in their simulations the last ∼6 h before observation being most relevant for particle growth. In this period, the ice particles typically travel 150–500 km southward. Before, the ice particles remained small ($< 20$ nm) for more than 60 h. In agreement with this, NLC above Kühlungsborn are generally observed during southward wind conditions (Gerding et al., 2007, 2013a, b), and also
in the Southern Hemisphere NLC even at high latitudes (69° S) are typically limited to equatorward winds (Morris et al., 2007; Klekociuk et al., 2008). In contrast, Stevens et al. (2017) found a dominating dependence of NLC on local temperatures even at midlatitudes, using NOGAPS-ALPHA assimilated model data. The authors explicitly neglected transport effects by using a 0-D NLC model, making a direct comparison with our findings difficult. Similarly, Herron et al. (2007) and Hultgren et al. (2011) found local effects dominating the formation of NLC for their midlatitude observations.

Furthermore, we cannot exclude longitudinal differences for the formation of ice clouds. According to SOFIE observations our site is at the longitude of minimal NLC occurrence rates (Hervig et al., 2016), suggesting different cloud formation mechanisms compared to other longitudes. It has been shown before for our site that temperatures below the frost point as well as southward winds are necessary but not sufficient criteria for NLC observations (Gerding et al., 2007). Supersaturated regions upstream of Kühlungsborn are needed to foster the nucleation process and to allow for the particles to grow to sufficient sizes (Fig. 8, lower left). About one third of the events examined here show southward winds also above the layer. The fact that these air parcels do not contain ice particles again confirms that southward winds are necessary but not sufficient for ice existence above our site. On the other hand, in another third of our events we found northward winds directly above the cloud, i.e. advecting air from (presumably) warmer regions. We would like to note that the OSWIN radar often detects MSE in thin layers above 85 km altitude (e.g. Zecha et al., 2003). We cannot exclude that these MSE are formed by ice particles nucleated closeby. But these particles typically do not reside long enough at our site to grow to sizes that allow optical detection, or the supersaturated height range is too thin to allow effective growth during sedimentation. This is sketched in the lower right part of Fig. 8.

In contrast to the upper edges, the lower edges of NLC and PMSE at 69° N typically agree quite well (Kaifler et al., 2011), similar to our observations. This is in fact expected due to the fast sublimation of the ice particles with typically rising temperatures at the lower edge of the layer. Of course, the instrument's sensitivity needs to be taken into account for comparison of layer edges. For the lidar observations described here we set the threshold to $\beta = 0.3$, which is slightly smaller than the threshold used in Gerding et al. (2013b) for the same lidar ($\beta = 0.5$). We confirmed by manual inspection that the edges of the individual NLC are correctly identified and not affected by background noise. For the radar data we processed the data in units of SNR. The threshold is set to -12 dB to be above the typical noise limit of the radar. We tested the influence of this threshold by setting it to larger and smaller numbers. Figure 9 shows the same histogram as Fig. 5, but for thresholds of -10 dB **Fig.9** and -14 dB. As expected, the mean $z_{\mathrm{MSE}}^{\mathrm{up}}$ rises to 84.6 km if also weaker MSE are included. Limiting the data to $>$-10 dB, we found the mean $z_{\mathrm{MSE}}^{\mathrm{up}}$ at 84.4 km. In other words, the mean shift between both test scenarios is only 200 m, i.e. less than one altitude bin of the radar (300 m). Changing the NLC threshold has similar results. Also here the gradients at the layer edges are very large (see, e.g., Fig. 2) and a change of the threshold to, e.g., $\beta = 0.5$ would result in only minor changes of the histograms. Therefore, there are only minor effects of the thresholds on layer edges, while the layer maximum is not affected at all. In any case, the changes due to threshold adaptions are much smaller than the difference to observations at higher latitudes.

## 6 Summary and Conclusions

In this study we compared NLC and MSE altitudes from simultaneous observations at our midlatitude site Kühlungsborn (54° N, 12° E). There is general agreement that both phenomena represent ice clouds, with the visible echoes (NLC) being formed by ice particles of typically some tens of nanometers diameter. The radar echoes (MSE) can also be formed by smaller particles, but require a sufficient density of free electrons as well as structures in the plasma. We have presented examples for NLC-only and MSE-only periods as well as for simultaneous layers with at least partial overlap in the covered altitude region.

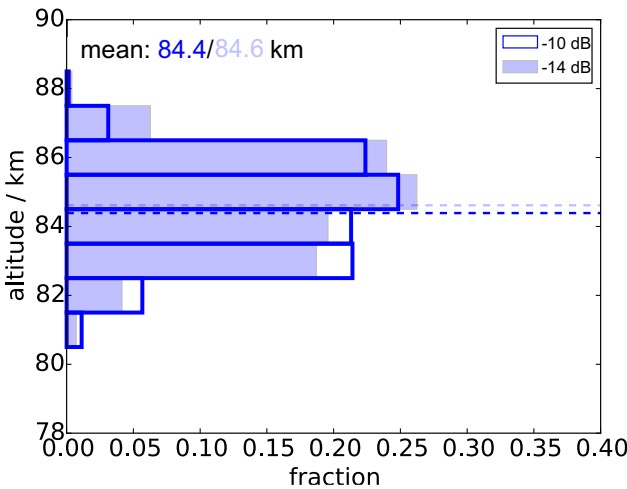

**Figure 9.** Same as Figure 5, but with different MSE thresholds. blue, open: MSE threshold SNR = -10 dB; light blue filled: MSE threshold SNR = -14 dB.

For the average layer parameters we only concentrated on simultaneous detections, i.e. we discarded nighttime NLC because of typically too small electron densities to form MSE. Furthermore we discarded all those high and weak MSE that are not accompanied with NLC. Overall, we got ~67 h of NLC/MSE data within the summers 2010 – 2016.

On average, the lower edges of NLC and MSE are identical, which is in agreement with the general understanding of quickly
sublimating ice particles at the bottom of the layer. The average values for the layer peaks and for the upper edges differ by 0.3 km and 0.5 km, respectively, with the MSE being slightly above the NLC. This comparatively small difference is in contrast to the observations at polar latitudes, where the PMSE often extend several kilometers above the NLC. We found that the ice cloud itself is much thinner compared to polar latitudes (under the assumption that the MSE layer thickness at midlatitudes is not limited by smaller electron density or missing turbulence). Clouds that already exist long enough to form large particles
(NLC) show only a thin layer of small particles (invisible for the lidar but visible as MSE) above the NLC at our site. Or they show no particles at all above the NLC, i.e. the upper edges of NLC and MSE coincide. Using simultaneous resonance lidar temperature soundings we typically found the atmosphere above the layer being too warm for ice existence, limiting the potential extent of the cloud. Meridional winds above the NLC do not show a preferential direction for the examined events. Altogether, these observations give evidence that local formation of MSE is possible, but these ice particles do not stay long
enough to grow to optically visible sizes. All layers that are observed as NLC are already formed and then advected to our site by the meridional wind. During advection and descent, the smaller particles grow to sizes of some ten nanometers. By this, they become detectable by lidar. This formation process must be taken into account if, e.g., midlatitude NLC observations are used to study trends or climate change in the mesopause region.

*Competing interests.* The authors declare that they have no conflict of interest.

*Acknowledgements.* We thank our colleagues Torsten Köpnick, Reik Ostermann, Michael Priester and Jörg Trautner for their support in lidar and radar operation and maintenance. We acknowledge the contributions of numerous students helping with continuous lidar soundings. Part of this work was supported by the Deutsche Forschungsgemeinschaft (DFG) under grant GE 1625/2-1. We thank the reviewers for their valu-
5    able comments that improved the manuscript. Presented data are available at ftp://ftp.iap-kborn.de/data-in-publications/ GerdingACP2018/.

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
