# Peer review of "Simultaneous observations of NLC and MSE at midlatitudes: Implications for formation and advection of ice particles"

_Atmospheric Chemistry and Physics, 2018_

## Referee Comment (RC1) · Anonymous Referee #1 · 24 Jul 2018

General comments

The title describes well what is shown in this manuscript, so I will not try to formulate it better. The authors use a large dataset from collocated, simultaneous radar measurements of MSE and lidar measurements of NLC. They select cases when MSE and NLC were present at the same time to characterize these two different, but related middle atmosphere phenomena. In the Introduction and the Discussion, the authors summarize well our present knowledge of the phenomena, and they show where their findings agree with existing knowledge, and where they can add new knowledge. The figures are excellent, clear and well described. The text is generally formulated clearly. The

authors describe well how they selected and analyzed the data.

Specific comments

What is new knowledge in this manuscript? - This can be found in the second paragraph of section 6 Summary and Conclusions, the first 2/3 of that paragraph. Section 5 Discussion and the excellent Figure 7 explain what this means, and why the differences between the combination PMSE/NLC at high latitudes and MSE/NLC at intermediate latitudes are as observed. I can imagine Figure 7 being used in lectures and review talks in the future.

The mechanism that creates MSE or PMSE is a complex one involving the existence of aerosol particles (ice particles), turbulence and the presence of free electrons and ions - three ingredients. The authors describe this well and with sufficient detail for this manuscript on lines 4/5 on page 13 in the Summary and conclusions. At other places in the text, the authors understandably shorten this already brief description, for instance on p. 2 l. 12, p. 2 l. 25, p. 8 l. 1, p. 10 l. 3, and p. 12 l. 1. They mention only the fact that particles must be present, but not the other two "ingredients", except on p. 10 l. 3, where they add turbulence but omit free electrons and ions. The authors, this reviewer, and many readers of the published paper know that all three ingredients are necessary, but scientists new to MSE and PMSE most likely do not. They may learn "small ice particles make MSE or PMSE", which is not a true statement. It would be clumsy to repeat the sentence from p. 13 l. 4/5 every time. Therefore I do not know an easy solution to the problem that I am trying to point out. As a tentative suggestion, the instance on p. 2 l. 12 could be formulated like this: "... (NLC), while radar echoes ((P)MSE) require ice particles, large or small, where smaller particles may be freshly formed in the ...". The word "require" seems to include the meaning that something else is required, too (turbulence and ionization). For the case of p. 8 l. 1, my tentative suggestion might be a small addition: "... visible for radars (signal strength proportional r^2, if turbulence and ionization in addition allow )." For p. 10 l. 3, my tentative suggestion is: "... turbulence is needed to create radar echoes

(Rapp and Lübken, 2004), as well as sufficient ionization. I ask the authors to kindly understand this paragraph of mine as a suggestion, no more. I leave it to them to find brief but complete solutions for the other instances that I have pointed out earlier in this paragraph.

Technical corrections

"to allow" is a verb with a distinctly different meaning than "to allow for", see Oxford, Cambridge, or Webster dictionaries. On p. 1 l. 4 (twice) and p. 5 l. 5, it seems "to allow" is what is really meant. p. 1 l. 17 "... respectively, have been performed since..." p. 1 l. 19 and 20: Consider "equatorward" instead of "south" to make the statement global. p. 2 l 10/11: The subject and the verb of this sentence do not agree logically. I suggest "give additional information", which is better, or perhaps "From simultaneous observations by lidar and radar, we gain additional..." p. 2 l. 15 The verb "to sediment" is intransitive. Therefore I recommend to delete the "been". p. 2 l.18/19: Consider adding "During darkness and outside the auroral oval, the ionization..." p. 3 l. 1: Replace "like" with "such as" p. 3 l. 19 I recommend "is achieved by a narrow field of view of the telescope". "Field-of-view" with dashes is an adjective. Without dashes, it is a noun. p. 3. l. 33 Here, I would recommend adding a dash in "phased-array" because "phased" and "array" belong together, but "phased" is not a qualifier for "field". p. 4 l. 2 and l. 8: "For receiving, ..." (comma) p. 4 l. 12 might be better formulated like this: "As we do not have an absolute calibration of the radar, we use SNR as an approximation for the echo intensity..." p. 5 l. 18 "The MSE quickly grew..."; p. 13 l. 10 "understanding of quickly sublimating..." . "Fast" does not form an adverb by adding "-ly". Correct the "grew" as well. p. 5 l. 26, p. 13 l. 8 and elsewhere: Just a comment from my side: Usually in everyday life and in laboratory physics, "high" is often used as synonymous with "large" and "low" as synonymous with "small". Here is a case where it becomes ambiguous, because we are writing about the atmosphere: Is the ionization too small, or is it too low in altitude? I know the former is meant, but there is a slight ambiguity. p. 6 l. 5 "1 km bins" (no dashes) p. 6 l. 7 lowercase "figure", as it is not a name

in this case. p. 11 l. 1 and l. 13: "... in our observations suggest that the layer of only small particles..." ; "(Hervig et al., 2016), suggesting different cloud formation mechanisms..." p. 13 l. 18 "extent" p. 13 l. 21 "descent" I do not understand the very last sentence with the verb "acknowledged". Perhaps "This formation process must be taken into account..." is meant or "This formation process must be allowed for" (here in the correct meaning of this verb).

---

## Referee Comment (RC2) · Anonymous Referee #2 · 26 Jul 2018

The authors combine two datasets from co-located instruments. A Rayleigh lidar observes NLC which is a direct measure of ice particles in the mesopause region, while a VHF radar observes mesospheric summer echoes which are by complicated physics linked to the presence of ice particles as well. Both phenomena are known to be closely related from detailed studies at polar latitudes. Both datasets from a mid-latitude site used here by the authors were described in detail before, so no new data is presented. As there is scientific interest regarding the occurrence of NLC at mid-latitudes, it seems nevertheless worthwhile to undertake this combination of the datasets.

However, the study presented here is not as extensive as the studies at polar latitudes.

[Figure]

Basically it is reduced to the comparison of three layer parameters: the lower and the upper layer edge of simultaneous NLC/MSE and their centroid altitude. The only relevant result of this study is a difference of 500 m between the upper edges, which differs significantly from 3.3 km found at polar latitudes. Even the authors do not seem to be surprised, though. They attribute it to reduced thickness of the MSE layer – but it is not clear if this has been shown before, and they don't think it is necessary to show it here as well.

Their only conclusion from the study is that advection is the main process for NLC occurrence at the observation site. This is by no means a new conclusion. From Gerding, JGR, 2007: "We conclude that NLC at midlatitudes are strongly coupled to the advection of preexisting ice particles from northern latitudes." and Gerding, GRL, 2013 "Comparing NLCs and ambient winds, we find strong indications for the meridional wind (advection) being the main driver for NLC occurrence above our site."

The authors claim to undertake the first statistical study at mid-latitudes. I acknowledge that this is a difficult task, and with their instrumentation they are also the only ones able to do this. The reason is that the NLC occurrence frequency is low, so with a lot of effort, only a very limited dataset is to be gained. At the same time, this makes these measurements highly valuable, and they should be treated accordingly. I fear that with 64 or 67 hours of data available to this study, it does not qualify to being statistical, or to being representative for NLC. The authors are not clear about the number of events or the number of independent profiles, but there is reason to suspect these numbers are low. There is one flaw in their discussion regarding the mean centroid altitude compared to their previous NLC statistics. I think they either made a mistake or the dataset cannot be considered to be representative. A large percentage of the NLC dataset (two third) was not included in this study, which is sad, and the reason wasn't explicated in sufficient detail.

In my impression the potential of the data shown was not fully exploited. The authors dedicate one section to the display of four cases with varied, sometimes intriguing

morphology, but no physical explanation is offered. It is therefore not clear why they are shown at all. The following statistics of lower edges makes the reader wonder if the morphology with a double layer is correctly represented. Especially the extreme cases of the statistics would be worth taking a closer look at, e.g. when the MSE lower edge is located 3 km below the NLC lower edge. I also doubt that the statistics of lower and upper edges result in the same correlation coefficient, as they look different to me.

Another critisicm is that the authors invoke an incorrectly simplified image of PMSE physics in particular, by stating that NLC are created by large ice particles and MSE are created by small ice particles, or even simpler, that lidar observes large and radar observes small particles. Here and there some references to our understanding of the physics of PMSE are interspersed, mostly when some explanation for some discrepancy is needed.

Questions and comments regarding science are following sorted by line numbers. A second set of comments with more technical corrections is appended.

p. 1, l. 1 This is the very first sentence, and it is not very precise: radar measurements are not a direct observation of ice particles, you shouldn't make such a statement in the very beginning. And they can also be observed optically by eye or camera. And why the focus on ground-based observations here? Its not yet clear what you are after.

p. 1, l. 2 Second sentence: that's too much of a simplification, reality is more complex

p. 1, l. 4 "allows for some insight" – yes, but that's now a very complicated task

p. 1, l. 5 I feel the need to object to the "statistical study". It is only 67 hours of data. It is more a compilation of cases, but not statistics.

p. 1, l. 6 MSE is not a direct measurement of ice clouds

p. 1, l. 18 and from space. You mention "stations", which I read as ground-based, but then cite results from satellite observations in the second sentence, so it's worth being included in the first sentence that there are also satellite observations.

p. 1, l. 22 this might be the most suitable place to explain in necessary detail the relation between lidar-observed NLC and radar-observed (P)MSE, and not only give the citation. The differences are not restricted to occurrence and vertical extension, but the physical mechanisms are very different. As you only give this information piece by piece throughout the manuscript, you might want to take the chance to make this very clear here. Then the reader won't be misleaded and then be surprised while reading that it's in fact more complicated then you had hinted.

p. 2, l. 12 already here it would be useful to have the physics of PMSE explained

p. 2, l. 14 that's not very obvious. It could have been created within the NLC layer for all we know

p. 2, l. 25 equating "local ice formation" with "observations of PMSE" is too much of a simplification

p. 3, l. 16 you should motivate why the diurnal variation of NLC is of relevance for this paper if you cite it

p. 3, l. 23 do you not normalize to density? I thought the common technique is to normalize to density and then take the ratio of the Mie scatter to this?

p. 3, l. 27 i.e. smoothed with 15 min width?

p. 3, l. 32 now you proceed with the radar. I suggest subsections per instrument. You started the paragraph by mentioning the commissioning of the instrument in summer 2010, and with no word you give any numbers on observations statistics!

p. 5, l. 1 There is a break here. There was a description of the radar dataset and then, with no subsection change, the text continues with different types of agreement between the observations

Fig. 2 it might help the reader to indicate times with solar elevation above 5 deg, as it seems to be important to PMSE occurrence

p. 5, l. 7 "often filled the same volume" the expression is not elegant, it's not very precise and it's not even true when I look at the figure!

p. 5, l. 12 the observation of MSE is not a detection of ice particles, once again

p. 5, l. 23 especially Fig. 2d seems to be a case with lots of features sparking many questions regarding the physics. No explanation is given! That's a bit frustrating to the reader.

p. 5, l. 23 Again on this paragraph, it is not clear what the intention is. You want to show four cases to make what point? That you also see features that others have described? It is not comprehensive, there is no explanation given, no conclusion is drawn, so why? You show layers with intricate morphology, but you do not do justice to this. In the following you restrict yourselves to three parameters only.

p. 5, l. 27 MSE that are too high to be observed by lidar? Surely there is no limit at e.g. 85 km for the lidar? And MSE that are too weak to be observed by lidar? They are not observed by lidar in any case.

p. 5, l. 30 might be worth giving an update on the occurrence rate: 188.5 h / 3337 h is $\sim$ 5 %. And is 3337 hours the "operation time" or the time with high-quality data suitable for NLC detection? Cause that might be significantly lower than the operation time. And it is only this that is relevant information for scientific purposes, the former is of interest to the laser engineer only.

p. 5, l. 29 I am surprised by the low number of 67 hours. You are throwing away 64 % of your precious, rare data on NLC. Might be worth to state why: So many hours due to solar elevation below 5 deg, so many hours due to missing PMSE at night, so many hours due to radar downtime

p. 5, l. 32 it makes you wonder if the study is representative for NLC, if you only use 36 % of the data... Fig. 1, 2 the five events shown amount to 17 hours out of the 67 hours. So I extrapolate that your statistics is based on 20 events? You withhold that

number, but you should give it

p. 6, l. 4 as shown in Fig. 1, but what about the multiple layers in Fig. 2d? These are several hours at least. In a dataset this small, it would be worth taking very good care of this.

p. 6, l. 4 1931 profiles a 2 minutes are 64 hours. But you said the NLC data was smoothed with 15 min running mean, so only 256 profiles are independent, aren't they, and not 1931?

p. 6, l. 7 82.6 km for the lower edge seems quite high, how does this compare to polar latitudes? This is 82.1 km, I checked, so you might want to discuss this

p. 6, l. 14 any physical explanation for the 4-5 km difference?

p. 6, l. 15 "can also be explained" and what was the first explanation if this is the second? The "morning twilight" is no obvious physical explanation

Fig. 3b there are MSE altitudes 3 km below the NLC altitude, you didn't mention this

Fig. 5b I can't believe that this distribution has the same correlation coefficient as the one in Fig. 3b. Can you check this number again?

p. 8, l. 1 no ice particles are visible for radars

p. 9, l. 4 so this is evidence for local formation of ice clouds then?

p. 9, l. 9 "as expected" you should state the observations and then draw conclusions, and not expect something

p. 9, l. 16 atmospheric conditions like haze and solar background are the same to the two lidars, so they can't be the reason for a smaller dataset in one? Either it's a technical limitation or operational?

p. 9, l. 17 seven events are how many independent profiles?

p. 10, l. 12 as you showed, multi-year is not enough to be either statistical or representative

p. 10, l. 15 The mean peak altitude of this study is 83.3 and not 82.6 km. This was the mean lower edge. So this does not compare at all to the centroid altitude statistics and must be explained. Either you made a mistake, or this study is not representative at all.

p. 10, l. 24 and the lower edge in Kaifler et al. (2011) is 82.1 km, which is 500 m below your results

p. 10, l. 30 you didn't evaluate the thickness of the PMSE layer, so you need to cite for this statement

p. 11, l. 5 is this a result of Kiliani et al. (2013)? 150 km is not a large distance at all, I'd be surprised

p. 12, l. 12 if -14 dB gives similar results than -12 dB, then -12 dB is not the noise limit, or am I wrong?

p. 13, l. 5 here, in the conclusions, this is the first time that structures in the plasma are mentioned

Technical corrections:

p. 1, l. 8 Please don't italicize indices (low, NLC, MSE, I mean: typeset with $z_\mathrm{NLC}$ in LaTeX)

p. 1, l. 10 expression: "typically do not expand much above". (expression: ".." in the following always means that I feel the language could be improved here)

p. 2, l. 2 expression: "indicator for temperatures being below the frost point"

p. 2, l. 4 "we utilize"

p. 2, l. 6 expression: "particular important"

p. 2, l. 6 "partly used" that might be an unfortunate expression. You might mean all

kind of things.

p. 2, l. 10 the observations do not gain additional information

p. 2, l. 16 expression: "observations to examine this question"

p. 2, l. 16 delete "obviously"

p. 2, l. 24 expression: "extend several kilometers higher"

p. 3, l. 11 expression: "observations are performed"

p. 3, l. 15 you already noted in line 11 that it is daylight-capable

p. 3, l. 19 "of ∼60 murad", you already mentioned that it is narrow

p. 3, l. 22 Noctilucent Clouds -> NLC

p. 3, l. 22 remove "in the NLC altitude "

p. 3, l. 30 "evaluated manually"

p. 3, l. 31 "identified by software" you mean by some algorithm, which could be described here, or not

p. 4, l. 2 "For reception"

p. 4, l. 3 please spell 6 as six, 7 as seven, throughout the manuscript

p. 4, l. 4 expression: "Time series resulted in length of 34.1 s"

p. 4, l. 5 expression: "the available time resolution for observations amounted to 2 min"

p. 4, l. 12 expression: "Due to the not available absolute calibration"

p. 5, l. 1 expression: "different types of agreement" that could be phrased somehow better

p. 5, l. 2 if it is the first or last event or one in between doesn't matter, I think

p. 5, l. 6 you might want to start a new paragraph for the discussion of each case

p. 5, l. 18 growed to -> grew into? Or maybe: developed into

p. 5, l. 20 expression: "slightly after each other"

p. 5, l. 23 This paragraph starting at p. 5, l. 1 should be put into a separate subsection with paragraphs

p. 7, l. 1 expression "more pointlike"

p. 7, l. 4 delete blank between 4 and .

p. 7, l. 6 (Fig. 4, right)

p. 8, l. 1 "regions extends" one s is too much

p. 8, l. 2 "getting finally visible for lidars"

p. 9, l. 2 "new ice layer" well, "new" in what sense, maybe "another"?

p. 10, l. 10 observation probability == occurrence frequency?

p. 10, l. 13 "the first RMR lidar" doesn't really matter here if it was the first?

p. 10, l. 31 descend -> descent, also p. 13, l. 21

p. 11, l. 1 expression: "hint to the conclusion"

p. 11, l. 1 expression: "the layer of only small particles"

p. 11, l. 16 to allow "for"

p. 12, l. 8 "which" is slightly smaller

p. 13, l. 18 extent

---

## Author Comment (AC1) · 29 Aug 2018

**Authors response on "Simultaneous observations of NLC and MSE at midlatitudes: Implications for formation and advection of ice particles" by Michael Gerding et al.**

**Anonymous Referee #1**

*We thank the reviewer for careful reading and the positive feedback. Answers to the specific comments are given below (in italics). New line numbers refer to the manuscript with marked changes.*

General comments

The title describes well what is shown in this manuscript, so I will not try to formulate it better. The authors use a large dataset from collocated, simultaneous radar measurements of MSE and lidar measurements of NLC. They select cases when MSE and NLC were present at the same time to characterize these two different, but related middle atmosphere phenomena. In the Introduction and the Discussion, the authors summarize well our present knowledge of the phenomena, and they show where their findings agree with existing knowledge, and where they can add new knowledge. The figures are excellent, clear and well described. The text is generally formulated clearly. The authors describe well how they selected and analyzed the data.

Specific comments

What is new knowledge in this manuscript? - This can be found in the second paragraph of section 6 Summary and Conclusions, the first 2/3 of that paragraph. Section 5 Discussion and the excellent Figure 7 explain what this means, and why the differences between the combination PMSE/NLC at high latitudes and MSE/NLC at intermediate latitudes are as observed. I can imagine Figure 7 being used in lectures and review talks in the future.

*We thank the reviewer for this comment. This is indeed the main conclusion of the paper, taking the limitations because of the complex origin of MSE into account.*

The mechanism that creates MSE or PMSE is a complex one involving the existence of aerosol particles (ice particles), turbulence and the presence of free electrons and ions - three ingredients. The authors describe this well and with sufficient detail for this manuscript on lines 4/5 on page 13 in the Summary and conclusions. At other places in the text, the authors understandably shorten this already brief description, for instance on p. 2 l. 12, p. 2 l. 25, p. 8 l. 1, p. 10 l. 3, and p. 12 l. 1. They mention only the fact that particles must be present, but not the other two "ingredients", except on p. 10 l. 3, where they add turbulence but omit free electrons and ions. The authors, this reviewer, and many readers of the published paper know that all three ingredients are necessary, but scientists new to MSE and PMSE most likely do not. They may learn "small ice particles make MSE or PMSE", which is not a true statement. It would be clumsy to repeat the sentence from p. 13 l. 4/5 every time. Therefore I do not know an easy solution to the problem that I am trying to point out.

*We apologize for explaining the complex origin of MSE and the limitations for the interpretation of the signal too late in the manuscript. Following also the suggestions of Referee #2, we added a short description of the origin of MSE in the beginning of Section 1 (p. 2 l. 6-8) and repeated the issue throughout the manuscript (e.g., p. 7 l. 1-5).*

As a tentative suggestion, the instance on p. 2 l. 12 could be formulated like this: "... (NLC), while radar echoes ((P)MSE) require ice particles, large or small, where smaller particles may be freshly formed in the ...". The word "require" seems to include the meaning that something else is required, too (turbulence and ionization).

*Many thanks. Inspired by this formulation we changed the phrasing to "… radar echoes ((P)MSE), ionization and turbulence provided, indicate small or large ice particles, where smaller particles may be freshly formed in the …" (p. 2 l. 21).*

For the case of p. 8 l. 1, my tentative suggestion might be a small addition: "... visible for radars (signal strength proportional r^2, if turbulence and ionization in addition allow )."

*We rephrased to "… if turbulence and ionization allow)" (p. 9 l. 3).*

For p. 10 l. 3, my tentative suggestion is: "... turbulence is needed to create radar echoes (Rapp and Lübken, 2004), as well as sufficient ionization.

*We want to make another point here and rephrased: "and continuous turbulence is needed to create radar echoes in contrast to intermittent turbulence being sufficient in combination with larger ice particles (Rapp and Lübken, 2004). Additional ionization is needed in both cases." (p. 11 l. 17/18)*

I ask the authors to kindly understand this paragraph of mine as a suggestion, no more. I leave it to them to find brief but complete solutions for the other instances that I have pointed out earlier in this paragraph.

*We hope, the complex issue of (P)MSE and the influence on the interpretation are much better described now.*

Technical corrections

"to allow" is a verb with a distinctly different meaning than "to allow for", see Oxford, Cambridge, or Webster dictionaries. On p. 1 l. 4 (twice) and p. 5 l. 5, it seems "to allow" is what is really meant.

*Rephrased or corrected*

p. 1 l. 17 "... respectively, have been performed since..."

*Changed (cf. Reviewer #2)*

p. 1 l. 19 and 20: Consider "equatorward" instead of "south" to make the statement global.

*Done*

p. 2 l 10/11: The subject and the verb of this sentence do not agree logically. I suggest "give additional information", which is better, or perhaps "From simultaneous observations by lidar and radar, we gain additional..."

*Done (p. 2 l. 18)*

p. 2 l. 15 The verb "to sediment" is intransitive. Therefore I recommend to delete the "been".

*Corrected*

p. 2 l.18/19: Consider adding "During darkness and outside the auroral oval, the ionization..."

*Done*

p. 3 l. 1: Replace "like" with "such as"

*Done*

p. 3 l. 19 I recommend "is achieved by a narrow field of view of the telescope". "Field-of-view" with dashes is an adjective. Without dashes, it is a noun.

*Corrected*

p. 3. l. 33 Here, I would recommend adding a dash in "phased-array" because "phased" and "array" belong together, but "phased" is not a qualifier for "field".

*Corrected*

p. 4 l. 2 and l. 8: "For receiving, ..." (comma)

*Corrected*

p. 4 l. 12 might be better formulated like this: "As we do not have an absolute calibration of the radar, we use SNR as an approximation for the echo intensity..."

*Rephrased. Many thanks for the suggestion.*

p. 5 l. 18 "The MSE quickly grew...";

*Corrected*

p. 5 l. 26, p. 13 l. 8 and elsewhere: Just a comment from my side: Usually in everyday life and in laboratory physics, "high" is often used as synonymous with "large" and "low" as synonymous with "small". Here is a case where it becomes ambiguous, because we are writing about the atmosphere: Is the ionization too small, or is it too low in altitude? I know the former is meant, but there is a slight ambiguity.

*Checked throughout*

p. 6 l. 5 "1 km bins" (no dashes) p. 6 l. 7 lowercase "figure", as it is not a name in this case.

*Corrected*

p. 11 l. 1 and l. 13: "... in our observations suggest that the layer of only small particles..." ; "(Hervig et al., 2016), suggesting different cloud formation mechanisms..."

*Rephrased*

p. 13 l. 10 "understanding of quickly sublimating..." . "Fast" does not form an adverb by adding "-ly". Correct the "grew" as well.

*Corrected*

p. 13 l. 18 "extent"

*Corrected*

p. 13 l. 21 "descent"

*Corrected*

I do not understand the very last sentence with the verb "acknowledged". Perhaps "This formation process must be taken into account..." is meant or "This formation process must be allowed for" (here in the correct meaning of this verb).

*Yes, this was meant. Rephrased.*

---

## Author Comment (AC2) · 29 Aug 2018

**Authors response on "Simultaneous observations of NLC and MSE at midlatitudes: Implications for formation and advection of ice particles" by Michael Gerding et al.**

**Anonymous Referee #2**

> *We thank the reviewer for the careful reading and the helpful comments. Answers to the specific comments are given below (in italics). New line numbers refer to the manuscript with marked changes.*

The authors combine two datasets from co-located instruments. A Rayleigh lidar observes NLC which is a direct measure of ice particles in the mesopause region, while a VHF radar observes mesospheric summer echoes which are by complicated physics linked to the presence of ice particles as well. Both phenomena are known to be closely related from detailed studies at polar latitudes. Both datasets from a mid-latitude site used here by the authors were described in detail before, so no new data is presented. As there is scientific interest regarding the occurrence of NLC at mid-latitudes, it seems nevertheless worthwhile to undertake this combination of the datasets.

> *Many thanks for this comment. We truly estimate the combination of both data sets to be worthwhile. This has rarely done on a large data set before, and never for midlatitudes. We think to gain additional knowledge from this combination, even if the process for MSE is indeed complex and does not only involve ice particles.*

However, the study presented here is not as extensive as the studies at polar latitudes. Basically it is reduced to the comparison of three layer parameters: the lower and the upper layer edge of simultaneous NLC/MSE and their centroid altitude. The only relevant result of this study is a difference of 500 m between the upper edges, which differs significantly from 3.3 km found at polar latitudes. Even the authors do not seem to be surprised, though. They attribute it to reduced thickness of the MSE layer – but it is not clear if this has been shown before, and they don't think it is necessary to show it here as well.

> *The reviewer is right that the MSE data can also be described by their thickness that is on average lower compared to polar latitudes. For the comparison with the NLC layer we found it reasonable to differentiate between upper and lower edges because the layer thickness on its own does not say anything about the relation to NLC altitudes. Kaifler et al. (2011) were able to do a more detailed analysis based on a much larger data set from high NH latitudes, while Klekociuk et al. (2008) made an initial study on a smaller data set from high SH latitudes. Given our limited data set and the instrumental changes of the OSWIN radar we hesitate to do an extensive analysis like Kaifler et al. (2011). Edge altitudes are comparatively robust against instrumental changes, while, e.g., occurrence rates may not. Furthermore we wanted to focus on potential differences to higher latitudes, which we mainly found in the upper edges.*

> *We have tried to sharpen the description of the relevant results in different parts of the manuscript (see below).*

Their only conclusion from the study is that advection is the main process for NLC occurrence at the observation site. This is by no means a new conclusion. From Gerding, JGR, 2007: "We conclude that NLC at midlatitudes are strongly coupled to the advection of

preexisting ice particles from northern latitudes." and Gerding, GRL, 2013 "Comparing NLCs and ambient winds, we find strong indications for the meridional wind (advection) being the main driver for NLC occurrence above our site."

> *Yes, we have truly claimed this before. But there are other, partly newer publications that propose local processes. During the analysis of the NLC/MSE data we found this additional indication for advection, supporting our previous papers. We see this observation and conclusion as relevant, especially since these combined observations are only possible with our (still unique) combination of instruments.*

The authors claim to undertake the first statistical study at mid-latitudes. I acknowledge that this is a difficult task, and with their instrumentation they are also the only ones able to do this. The reason is that the NLC occurrence frequency is low, so with a lot of effort, only a very limited dataset is to be gained. At the same time, this makes these measurements highly valuable, and they should be treated accordingly. I fear that with 64 or 67 hours of data available to this study, it does not qualify to being statistical, or to being representative for NLC. The authors are not clear about the number of events or the number of independent profiles, but there is reason to suspect these numbers are low.

> *We are happy that the value of our observations is acknowledged. We apologize if the term "statistical analysis" is misleading here. We have replaced it in the revised version (e.g. "comparative" (p. 1 l. 3), "vertical distributions" (p. 3 l. 15), "on average" (p. 8 l. 13)). We now mention the number of 31 days with NLC/MSE in Section 2.4 (p. 7 l. 16). These events are representative for all NLC in terms of their altitude structure, as described in Section 5 (p. 11 l. 31 – p. 12 l. 6 , see also comment below). Of course, they may not in terms of brightness or diurnal variation. Furthermore, the MSE during simultaneous NLC are not representative for all MSE, as already described in Section 5. There are many high and/or weak MSE not represented here. But these MSE just support the presented conclusion about the formation processes.*

There is one flaw in their discussion regarding the mean centroid altitude compared to their previous NLC statistics. I think they either made a mistake or the dataset cannot be considered to be representative. A large percentage of the NLC dataset (two third) was not included in this study, which is sad, and the reason wasn't explicated in sufficient detail.

> *We apologize for the mistake about the mean NLC height in the beginning of Section 5. Indeed, the mean peak height here is 83.3 km. Many thanks for making us aware of this flaw in our discussion. We have revised and extended this section (p. 11 l. 31 – p. 12 l. 6), but we still state that the presented data are representative for all NLC. In the 2013a paper (data 1997-2011, nighttime only) we gave a number for the mean centroid altitude that is typically 0.2 km below the mean peak altitude. The mean peak altitude is easier to identify, and is 82.8 km for all NLC 2010 – 2016. The mean peak altitude for the selected days of simultaneous MSE is 83.0 km (mean centroid altitude 82.8 km). From this data set the very weak NLC profiles (beta < 0.3\*10[-10] /m/sr) are removed, e.g. during beginning and end of the event, as well as some profiles with very low NLC (80 km and below), because here the ionization is reduced and MSE are less likely.*

> *For this study we needed to remove NLC during nighttime (or low solar elevation, i.e. ionization) because of missing MSE. Furthermore we excluded the faintest NLC (beta < 0.3\*10[-10] /m/sr) because of typically bad SNR and therefore unreliable edge detection. We explain this in more detail in the revised manuscript (p. 7 l. 18-20).*

In my impression the potential of the data shown was not fully exploited. The authors dedicate one section to the display of four cases with varied, sometimes intriguing morphology, but no physical explanation is offered. It is therefore not clear why they are shown at all. The following statistics of lower edges makes the reader wonder if the morphology with a double layer is correctly represented. Especially the extreme cases of the statistics would be worth taking a closer look at, e.g. when the MSE lower edge is located 3 km below the NLC lower edge. I also doubt that the statistics of lower and upper edges result in the same correlation coefficient, as they look different to me.

> *We agree that the events presented in Figure 2 are worth further analysis, and we thank for the encouragement. Nevertheless, this is outside the scope of this paper. While showing very interesting dynamical structures, detailed analysis of these cases needs further information about ionization (electron densities) and turbulence. Wind data is available only with limited temporal and spatial resolution, while temperature data is completely missed for most cases. Figure 2 is presented to make the reader aware of this highly dynamic behavior of NLC and MSE – and the limitations for detection of layer edges from independent, asynchronous instruments. We have improved the description in the revised manuscript (p. 5 l. 20 and p. 6 l. 13 - p. 7 l. 8).*

> *Figure 2 shows some cases with larger differences of NLC and MSE edges. These are worth further analysis, but so far observations of electron density and turbulence are lacking at our site. Other large differences occur for technical reasons like different FOV sizes or asynchronous data.*

> *We have double-checked the correlation coefficients. The "outliers" are mainly single profiles, while the majority of events (dark color in Figures b) ) are along a line.*

Another critisicm is that the authors invoke an incorrectly simplified image of PMSE physics in particular, by stating that NLC are created by large ice particles and MSE are created by small ice particles, or even simpler, that lidar observes large and radar observes small particles. Here and there some references to our understanding of the physics of PMSE are interspersed, mostly when some explanation for some discrepancy is needed.

> *We thank the reviewer for making us aware of the insufficient explanation of MSE physics. We have added some explanation, e.g., in the beginning of the Introduction (p. 2 l. 6-8, see also below) and mention this topic also in the Abstract (p. 1 l. 5). The potential influence of the MSE physics on our results is, e.g., more clearly described in Sections 2.3 and 2.4, now (page 7).*

Questions and comments regarding science are following sorted by line numbers. A second set of comments with more technical corrections is appended.

p. 1, l. 1 This is the very first sentence, and it is not very precise: radar measurements are not a direct observation of ice particles, you shouldn't make such a statement in the very beginning. And they can also be observed optically by eye or camera. And why the focus on ground-based observations here? Its not yet clear what you are after.

> *We have changed the first sentence to "We have combined ground-based observations of ice particles in the summer mesopause by lidar (then often called Noctilucent Clouds, NLC) and radar (then called (Polar) Mesospheric Summer Echoes, (P)MSE) for a first comparative study on ice cloud altitudes at midlatitudes (Kühlungsborn/Germany, 54° N, 12° E)."*

p. 1, l. 2 Second sentence: that's too much of a simplification, reality is more complex

*We have added "but require sufficient ionization and turbulence at the ice cloud altitudes" at the end of the sentence.*

p. 1, l. 4 "allows for some insight" – yes, but that's now a very complicated task

*Phrase is changed to "…provides some rough information about …".*

p. 1, l. 5 I feel the need to object to the "statistical study". It is only 67 hours of data. It is more a compilation of cases, but not statistics.

*We have deleted "statistical", now saying "comparative study" (p. 1 l. 3) and avoid this term with respect to this study (cf. above).*

p. 1, l. 6 MSE is not a direct measurement of ice clouds

*Yes. Limitations are now mentioned, e.g., in the second sentence, in the Introduction and in Section 2.3. See also below.*

p. 1, l. 18 and from space. You mention "stations", which I read as ground-based, but then cite results from satellite observations in the second sentence, so it's worth being included in the first sentence that there are also satellite observations.

*We thank the reviewer for the careful reading. We have changed the sentence to "Noctilucent Clouds (NLC, also known as Polar Mesospheric Clouds, PMC) and Polar Mesospheric Summer Echoes (PMSE) are observed since several decades mainly in the polar regions by ground-based and space-based instruments as well as by human eye […]." (p. 1, l. 19-22)*

p. 1, l. 22 this might be the most suitable place to explain in necessary detail the relation between lidar-observed NLC and radar-observed (P)MSE, and not only give the citation. The differences are not restricted to occurrence and vertical extension, but the physical mechanisms are very different. As you only give this information piece by piece throughout the manuscript, you might want to take the chance to make this very clear here. Then the reader won't be misleaded and then be surprised while reading that it's in fact more complicated then you had hinted.

*We added a first sentence on the complex origin of PMSE here ("Later on it was revealed that PMSE additionally require sufficient ionization of the ambient air to get the ice particles charged and turbulence to produce plasma structures for scattering of the radar wave." p. 2 l. 6-8) and provide information about the implication of these requirements throughout the manuscript.*

p. 2, l. 12 already here it would be useful to have the physics of PMSE explained

*We added here "ionization and turbulence provided".*

p. 2, l. 14 that's not very obvious. It could have been created within the NLC layer for all we know

*The formation of ice particles from condensation nuclei happens most likely at the mesopause, i.e. above 85 km, where supersaturation is largest (e.g. Rapp and Thomas, 2006). At these altitudes NLC are extremely rare. Typical altitudes of NLC and MSE are mentioned on p. 2 l. 4/5.*

p. 2, l. 25 equating "local ice formation" with "observations of PMSE" is too much of a simplification

*We changed this part to "Li et al., JGR 2010, revealed from PMSE observations average ice particle radii being larger above 85 km than below. This can be explained by …" (p. 2 l. 34/35)*

p. 3, l. 16 you should motivate why the diurnal variation of NLC is of relevance for this paper if you cite it

*We deleted this part of the sentence.*

p. 3, l. 23 do you not normalize to density? I thought the common technique is to normalize to density and then take the ratio of the Mie scatter to this?

*The reviewer is right. We have added this information ("…from the aerosol backscatter normalized to the molecular backscatter, the molecular backscatter cross section, and a reference air density to quantify the cloud brightness", p. 4 l. 5/6).*

p. 3, l. 27 i.e. smoothed with 15 min width?

*Yes. We have changed the phrasing.*

p. 3, l. 32 now you proceed with the radar. I suggest subsections per instrument. You started the paragraph by mentioning the commissioning of the instrument in summer 2010, and with no word you give any numbers on observations statistics!

*Many thanks for the suggestion of subsections. The observations statistics is given in the last subsection 2.4, which is now introduced earlier in an overview sentence (p. 3 l. 23-25 and p. 7 l. 10-24).*

p. 5, l. 1 There is a break here. There was a description of the radar dataset and then, with no subsection change, the text continues with different types of agreement between the observations

*We made a new subsection here and added a short introduction (p. 5 l. 20).*

Fig. 2 it might help the reader to indicate times with solar elevation above 5 deg, as it seems to be important to PMSE occurrence

*Many thanks for this suggestion. Done.*

p. 5, l. 7 "often filled the same volume" the expression is not elegant, it's not very precise and it's not even true when I look at the figure!

*We changed the phrasing to "showed good agreement" and give more precise information thereafter (p. 5 l. 28-30).*

p. 5, l. 12 the observation of MSE is not a detection of ice particles, once again

*Ice particles are necessary for MSE. Therefore we can conclude from the presence of MSE to the existence of ice in the observed volume.*

p. 5, l. 23 especially Fig. 2d seems to be a case with lots of features sparking many questions regarding the physics. No explanation is given! That's a bit frustrating to the reader.

*We agree that this event is very interesting and has the potential for further studies. Unfortunately we have no electron density and turbulence measurements available to explain, e.g., the gap in the MSE at 3:30 UTC. Examination of gravity wave dynamics*

*would be very interesting, but is beyond our scope for this paper. To make the reader aware of the high potential of this event we added (p. 6 l. 13-15): "The variable structure of the ice layer with double layers indicates a highly dynamic behavior of the atmosphere with presence of strong gravity waves. Nevertheless, a detailed examination of the dynamical structure is beyond the scope of this paper."*

p. 5, l. 23 Again on this paragraph, it is not clear what the intention is. You want to show four cases to make what point? That you also see features that others have described? It is not comprehensive, there is no explanation given, no conclusion is drawn, so why? You show layers with intricate morphology, but you do not do justice to this. In the following you restrict yourselves to three parameters only.

*We are sorry for not describing the purpose of these examples. We added a new paragraph at the end of the subsection (p. 7 l. 1-8): "The examples shown above demonstrate the different relations of the NLC and MSE layer edges and the different degrees of accordance of the layers. This is in general agreement with observations at polar latitudes (e.g. Klekociuk et al., 2008; Kaifler et al., 2011). The examples indicate an often good concurrence of the lower edges but a worse agreement of the upper edges. If solar elevation (i.e. ionization) is sufficiently large, NLC are often but not always accompanied by MSE. The latter might be explained by missing turbulence, but this cannot be proven here because a lack of appropriate measurements. Periods with MSE but absent NLC can be caused by mainly small ice particles, resulting in lidar signals below the NLC detection threshold. In the following we neglect profiles of NLC without MSE as well as MSE without NLC to be sure that for this study all requirements for the observation of small and large ice particles are fulfilled (see below)."*

p. 5, l. 27 MSE that are too high to be observed by lidar? Surely there is no limit at e.g. 85 km for the lidar? And MSE that are too weak to be observed by lidar? They are not observed by lidar in any case.

*We deleted the ice cloud "too high" for lidar but left "too weak". The ice cloud may produce an optical signal below the threshold but is detected by the radar, e.g. in case of small particles. We add "detected as MSE" for clarity.*

p. 5, l. 30 might be worth giving an update on the occurrence rate: 188.5 h / 3337 h is ~ 5 %. And is 3337 hours the "operation time" or the time with high-quality data suitable for NLC detection? Cause that might be significantly lower than the operation time. And it is only this that is relevant information for scientific purposes, the former is of interest to the laser engineer only.

*We prefer not to mention the occurrence rate of NLC in general, as only NLC accompanied by MSE are used here. The 3337 h are the number of hours suitable for detection of NLC with ß>0.3. We added the term "usable" for clarity (p. 7 l. 20).*

p. 5, l. 29 I am surprised by the low number of 67 hours. You are throwing away 64 % of your precious, rare data on NLC. Might be worth to state why: So many hours due to solar elevation below 5 deg, so many hours due to missing PMSE at night, so many hours due to radar downtime

*We do not distinguish why the data are not used here, but have added a short list of reasons (p. 7 l. 18/19). Indeed, we would be happy if we could use more of the rare NLC data here, but at our site many NLC profiles either show a quite low backscatter*

*coefficient (20-30% with ß<0.3, estimated from Fig. 4 of Gerding et al., JGR, 2013) or appear during nighttime, when the ionization is typically too small to support MSE (cf. 8.5 h of solar elevation below 5° on 21 June).*

p. 5, l. 32 it makes you wonder if the study is representative for NLC, if you only use 36 % of the data. . . Fig. 1, 2 the five events shown amount to 17 hours out of the 67 hours. So I extrapolate that your statistics is based on 20 events? You withhold that number, but you should give it

*As mentioned above, we do not see a significant difference between the layer parameters of the NLC used here and of all NLC. We discuss this in more detail in Section 5. Therefore we consider our results representative for all NLC.*

*We added on p. 7 l. 16-18: "These data are distributed across 31~days with an average ice cloud duration of 2.2 h. For this study it is not relevant whether the ice observation is uninterrupted in time or not, because the layer parameters are derived based on individual (but smoothed) profiles."*

p. 6, l. 4 as shown in Fig. 1, but what about the multiple layers in Fig. 2d? These are several hours at least. In a dataset this small, it would be worth taking very good care of this.

*We added on p. 7 l. 29 "In the rare case of a double layer we take the lower edge of the lower layer and the upper edge of the upper layer together with the absolute maximum."*

p. 6, l. 4 1931 profiles a 2 minutes are 64 hours. But you said the NLC data was smoothed with 15 min running mean, so only 256 profiles are independent, aren't they, and not 1931?

*The reviewer is right, not all profiles are independent. We added on p. 7 l. 30/31: "…, even if the respective smoothing needs to be taken into account for interpretation."*

p. 6, l. 7 82.6 km for the lower edge seems quite high, how does this compare to polar latitudes? This is 82.1 km, I checked, so you might want to discuss this

*The ice layer altitude is on average increasing with latitude, which is related to the changing temperature profile (smaller likelihood for supersaturation at, e.g., 82 km at 54°N compared to 70°N). Following the suggestion, we have added a short section in the discussion (p. 12 l. 15-18): "In their Table 3 they report also quasi-identical lower edges of NLC and PMSE, even if the $z^{low}$ at higher latitudes are observed 0.5 km above the midlatitude values. This latitudinal difference of $z^{low}$ can be explained by the general increase of NLC altitudes with latitude (Lübken et al., GRL, 2008; Chu et al., GRL, 2011) which is related to the ambient temperature structure."*

p. 6, l. 14 any physical explanation for the 4-5 km difference?

*We already tried to provide an explanation in the following sentences.*

p. 6, l. 15 "can also be explained" and what was the first explanation if this is the second? The "morning twilight" is no obvious physical explanation

*We rephrased the sentence before to make clear that this is a first explanation (p. 8 l. 8/9: "… in cases of MSE onset in the morning twilight where sometimes the MSE only agrees with the uppermost part (i.e. largest ionization) of the ice layer").*

Fig. 3b there are MSE altitudes 3 km below the NLC altitude, you didn't mention this

*We now expanded the explanation of the few larger altitude differences by "Rarely, the different size dependency of lidar and radar signals can lead to MSE even a few km below the NLC." (p. 8 l. 12/13)*

Fig. 5b I can't believe that this distribution has the same correlation coefficient as the one in Fig. 3b. Can you check this number again?

*We double-checked this number without finding an error.*

p. 8, l. 1 no ice particles are visible for radars

*We changed the phrasing to "detected by".*

p. 9, l. 4 so this is evidence for local formation of ice clouds then?

*Potentially, but a final proof cannot be given from the available data.*

p. 9, l. 9 "as expected" you should state the observations and then draw conclusions, and not expect something

*We added "from previous observations" (p. 11, l. 1). We want to make clear that this is not the first observation of southward wind during NLC/MSE.*

p. 9, l. 16 atmospheric conditions like haze and solar background are the same to the two lidars, so they can't be the reason for a smaller dataset in one? Either it's a technical limitation or operational?

*The potassium lidar at 770 nm suffers more from hazy conditions than the RMR lidar at 532 nm due to enhanced scatter of the longwave fraction of solar radiation. We added "at near-infrared wavelengths" (p. 11 l. 8).*

p. 9, l. 17 seven events are how many independent profiles?

*The seven events cover more than 25 h of data, but this number includes also periods during night/twilight, when the MSE has not set in, yet. The temperature data set for these days is much larger because it is not limited to NLC. On the other hand, temperatures have been calculated every 15 min with 2 h integration. We hesitate to provide all these numbers in the paper. We do not observe that the ambient conditions change drastically within the individual events, i.e. an event-wise classification is justified.*

p. 10, l. 12 as you showed, multi-year is not enough to be either statistical or representative

*We replaced "statistical analysis" by "analyses of average layer parameters".*

p. 10, l. 15 The mean peak altitude of this study is 83.3 and not 82.6 km. This was the mean lower edge. So this does not compare at all to the centroid altitude statistics and must be explained. Either you made a mistake, or this study is not representative at all.

*We are sorry for this mistake and thank the reviewer for his careful reading. As described above the selected cases are still representative for NLC in general. We have corrected the numbers and explain these now in more detail.*

p. 10, l. 24 and the lower edge in Kaifler et al. (2011) is 82.1 km, which is 500 m below your results

*As mentioned above, we have added and explained this difference in the Discussion (p. 12 l. 16-18).*

p. 10, l. 30 you didn't evaluate the thickness of the PMSE layer, so you need to cite for this statement

*We have added a reference (Kaifler et al., 2011).*

p. 11, l. 5 is this a result of Kiliani et al. (2013)? 150 km is not a large distance at all, I'd be surprised

*This is a result of Kiliani et al. and relates to a mean wind speed of 7 m/s (or 23 m/s for their upper limit). However, we see the old phrasing potentially misleading and changed it to (p. 12 l. 32/33): "In this period, the ice particles typically travel 150-500 km southward. Before, the ice particles remained small (< 20 nm) for more than 60 h."*

p. 12, l. 12 if -14 dB gives similar results than -12 dB, then -12 dB is not the noise limit, or am I wrong?

*We rephrased "The threshold is set to -12 dB based on the noise limit of the radar." to "The threshold is set to -12 dB to be above the typical noise limit of the radar." (p. 14 l. 5)*

p. 13, l. 5 here, in the conclusions, this is the first time that structures in the plasma are mentioned

*As mentioned in the above comments we now explain the complex origin of MSE much earlier. We thank the reviewer for making us aware of this.*

Technical corrections:

p. 1, l. 8 Please don't italicize indices (low, NLC, MSE, I mean: typeset with $z_\mathrm{NLC}$ in LaTeX)

*Done*

p. 1, l. 10 expression: "typically do not expand much above". (expression: ".." in the following always means that I feel the language could be improved here)

*Changed to "typically do not stretch much higher than the NLC" (p. 1 l. 13).*

p. 2, l. 2 expression: "indicator for temperatures being below the frost point"

*Deleted "being"*

p. 2, l. 4 "we utilize"

*Changed*

p. 2, l. 6 expression: "particular important"

*Changed to "in particular"*

p. 2, l. 6 "partly used" that might be an unfortunate expression. You might mean all kind of things.

*Deleted "partly"*

p. 2, l. 10 the observations do not gain additional information

*Changed to "give additional information" (cf. Reviewer #1)*

p. 2, l. 16 expression: "observations to examine this question"

*Changed to "solve this question"*

p. 2, l. 16 delete "obviously"

*Done*

p. 2, l. 24 expression: "extend several kilometers higher"

*Changed to "stretch several kilometers higher" (p. 2 l. 32)*

p. 3, l. 11 expression: "observations are performed"

*Changed to "made"*

p. 3, l. 15 you already noted in line 11 that it is daylight-capable

*We kept this but added "… and replaced the former RMR lidar used for nighttime NLC statistics." (p. 3 l. 27)*

p. 3, l. 19 "of _60 murad", you already mentioned that it is narrow

*Deleted "narrow"*

p. 3, l. 22 Noctilucent Clouds -> NLC

*Done*

p. 3, l. 22 remove "in the NLC altitude "

*Done*

p. 3, l. 30 "evaluated manually"

*Word order changed*

p. 3, l. 31 "identified by software" you mean by some algorithm, which could be described here, or not

*Changed to "by an algorithm"*

p. 4, l. 2 "For reception"

*Done*

p. 4, l. 3 please spell 6 as six, 7 as seven, throughout the manuscript

*Done*

p. 4, l. 4 expression: "Time series resulted in length of 34.1 s"

*Changed to "Time series of 1024 data points are acquired within 34.1 s." (p. 5 l. 8)*

p. 4, l. 5 expression: "the available time resolution for observations amounted to 2 min"

*Changed to "the time resolution for MSE observations is 2 min."*

p. 4, l. 12 expression: "Due to the not available absolute calibration"

*Changed to "As we do not have an absolute calibration of the radar, we use SNR as an approximation for the echo intensity." according to the suggestion of Reviewer #1 (p. 5 l. 17).*

p. 5, l. 1 expression: "different types of agreement" that could be phrased somehow better

*Rephrased to "Similar to previous studies we find partly very large agreement between NLC and MSE, while there are differences in other cases" (p. 5 l. 21/22)*

p. 5, l. 2 if it is the first or last event or one in between doesn't matter, I think

*Changed to "shows an events that was observed on 17 June 2010"*

p. 5, l. 6 you might want to start a new paragraph for the discussion of each case

*Done*

p. 5, l. 18 growed to -> grew into? Or maybe: developed into

*Changed to "grew into".*

p. 5, l. 20 expression: "slightly after each other"

*Deleted*

p. 5, l. 23 This paragraph starting at p. 5, l. 1 should be put into a separate subsection with paragraphs

*Done*

p. 7, l. 1 expression "more pointlike"

*Rephrased to "only ~1/1700 of this" (p. 8 l. 11)*

p. 7, l. 4 delete blank between 4 and .

*will be done late when the \marginpar command is removed*

p. 7, l. 6 (Fig. 4, right)

*Done*

p. 8, l. 1 "regions extends" one s is too much

*Corrected*

p. 8, l. 2 "getting finally visible for lidars"

*Rephrased to "and become …"*

p. 9, l. 2 "new ice layer" well, "new" in what sense, maybe "another"?

*Changed*

p. 10, l. 10 observation probability == occurrence frequency?

*Changed*

p. 10, l. 13 "the first RMR lidar" doesn't really matter here if it was the first?

*Changed to "previous" to make clear that it was not the lidar used for the current study*

p. 10, l. 31 descend -> descent, also p. 13, l. 21

*Corrected*

p. 11, l. 1 expression: "hint to the conclusion"

*Changed to "suggest" (cf. Reviewer #1)*

p. 11, l. 1 expression: "the layer of only small particles"

*Deleted "small"*

p. 11, l. 16 to allow "for"

*Corrected*

p. 12, l. 8 "which" is slightly smaller

*Corrected*

p. 13, l. 18 extent

*Corrected*

---

## Author Response (AR2)

**Authors response on review of revised version of "Simultaneous observations of NLC and MSE at midlatitudes: Implications for formation and advection of ice particles" by Michael Gerding et al.  Anonymous Referee #2**

*We thank the reviewer for taking the time and for providing detailed comments. Answers to the specific comments are given one-by-one below (in italics). New line numbers refer to the manuscript with marked changes.*

The manuscript has improved during revision, especially regarding the recognition of PMSE physics and explanation of the examples shown. Corrections or extensions were applied to the manuscript where necessary. Although the authors answered to all questions, I have subsequent questions or comments to their extensions, as explicated below. The study of NLC and MSE lower and upper edges certainly is of scientific significance (even if no substantial new concepts, ideas, methods or datasets are introduced in the manuscript), especially in comparison to the respective results at polar latitudes, as this might give indications for different formation processes.

*We are happy that the reviewer acknowledges the scientific significance of this study. We would like to note that the presented data set of 7 years of simultaneous NLC and MSE records at midlatitudes is unique and has not been described before. The method is by intention similar to the method of Kaifler et al., 2011, and takes limitations for our smaller data set into account.*

Yet this remains a difficult task and the authors tried their best to supply with temperature and wind data, but the results are inconclusive. The question is still open whether the dataset is large enough to derive significant results and allow for interpretation. At some places, analysis that could aid interpretation is lacking, or results are not properly acknowledged.

*We agree that the data set is still comparatively small and interpretation needs some care. Therefore, we limit our conclusions. For example, Kaifler et al. subdivided their (much larger) data set into various subcategories, where we only had one category. We still think that our data set supports our conclusions. We prove this statement in detail below.*

I also want to make aware that the manuscript does not conform to the journals data policy, requesting the used datasets to be publically available in an online repository and to be cited accordingly.

*We decided to make the data publically available on an ftp server. Reference is added in the Acknowledgments.*

Layer thickness: There are several references to layer thickness throughout the text, however it was not analyzed. It is expected to be thin, but e.g. Fig 2d shows a rather thick layer. As it is just a combination of two analyzed variables, $z\_up - z\_low$, the effort to calculate at least mean values for NLC and MSE should not be unreasonable. It would aid interpretation to know if the layers are thinner than at polar latitudes. The authors suspect that, but can easily derive it from data.

*So far, we feared that such a plot might add too few information. We are happy to present it now (new Figure 6). Mean layer thicknesses are 1.35 km (NLC) and 1.89 km (MSE), in agreement with the mean upper and lower edges. Especially the MSE layer*

*is much thinner than at high latitudes (mean ~5 km, Kaifler et al., 2011). Some profiles show a thickness of more than 4 km, but as already mentioned in the description of Fig. 2d, these are rare occasions. The description of the new Fig. 6 starts on page 9, line 12.*

I think another flaw of the previous type was introduced during revision. In p. 12, l. 16: Kaifler et al. (2011) list z_low as 82.1 km which is not 0.5 km above the mid-latitude value of 82.6 km, but below. Thus it can not be explained by the general increase of NLC altitude with latitude. With about 50 m per degree of latitude the z_low from high latitude of 82.1 km would shift down to 81.4 km, and not up to 82.6 km. Furthermore, this cited increase with latitude refers to the centroid altitude, and it is not clear at all what applies for lower edges. Again, some more intelligence on this topic might be revealed by the analysis of the layer thickness.

*We apologize for the confusion of high- and mid-latitude numbers and the, therefore, erroneous explanation. Indeed, the lower edge at high latitudes is 500 m below (!) the lower edge at mid-latitudes. To some extent, the good agreement between lower edges of NLC and MSE at both locations is related to the thresholds used for selection of the data. For ALOMAR, Kaifler et al. (2011) have set the MSE threshold to 5 dB relative signal power and the NLC threshold to ß>4 \*10$^{-10}$ /m/sr. For Kühlungsborn, we chose a MSE threshold of -12 dB SNR and a NLC threshold of ß>0.3\*10$^{-10}$ /m/sr, based on the noise limit of the data. From Fig. 5 of Kaifler et al., the altitude change of lower and upper edges depending on the threshold for data selection can be seen. For NLC, we find similar dependencies also for our site, i.e. increasing the brightness limit for our data set lowers the mean NLC lower edge by a few 100 m. This is because bright NLC are typically found at lower altitudes, as also the reviewer stated below.*

*We would like to point out that the data selection results in a systematic difference of lower edge altitudes of a few 100 m. The upper NLC edges are affected similarly, but the difference is smaller (found by Kaifler et al. as well as in our data set). The upper edge of only the strong MSE is indeed expected to be lower than the upper edge of all MSE. But this altitude change is still much smaller than the difference found between high and mid latitudes.*

*Please note that the event selection described above is based on the signal maxima. After the selection process in Kaifler et al. (2011), the particular edges are identified at a lower signal level. For Kühlungsborn the data selection threshold and the edge threshold are set the same. We have demonstrated in Fig. 9 (new number) that the mean edge height of MSE depends only weakly on the edge threshold.*

*We have changed the erroneous description of the lower edge comparison and provide a discussion of the edge altitudes and their threshold dependence (p. 13, l. 3-6).*

Related to this topic is another missing quantity which is z_up for the complete MSE dataset. It is very important for interpretation and should not be omitted. The authors hint that it could be significantly larger than 84.5 km which is z_up of NLC during MSE. This would mean that high-altitude MSE are suppressed during NLC conditions, which would be a major result. I don't think the authors interpretation is correct, they state that the small particles at high altitudes have grown and sedimented and are thus removed from high altitude during NLC conditions. But this process would not suppress subsequent reformation of local (!) high-altitude MSE, as is obviously observed during non-NLC conditions.

*In the figure below we compare the occurrence of MSE at a particular altitude bin for our data set (simultaneous with NLC) and for all MSE (as published by Zecha et al., 2003). Normalization is different, but it can be easily seen that the fraction of high-reaching MSE is larger in the data set without separation. (Dashed lines are drawn to guide the eye.) We hesitate to provide a number for z_up for the complete data set because it includes many hours of MSE without lidar observations, i.e. likely mixing conditions with and without NLC. The evaluation of the complete MSE data set is in fact out of the scope of this paper. The low MSE layer width indeed says that high-altitude ice clouds rarely exist together with the comparatively low NLC (cf. new Figure 6). The relevance of sedimentation is well established and, e.g., presented by Rapp and Thomas (2006) and Kiliani et al. (2013). At higher latitudes the reformation of local high altitude (P)MSE may occur, because the height range of supersaturation is typically much larger. At mid-latitudes we expect a thinner supersaturated region (e.g. Gerding et al., 2007), and therefore low-altitude NLC/MSE and high-altitude MSE rarely exist at the same time at a particular location. We tried to sketch this in Figure 8 (new number).*

[Figure]

The fact that most NLC had to be discarded due to missing MSE is another major result, which I think is different from polar latitudes. I understand that the authors refrain from analyzing occurrence frequencies for good reason (limited statistics), but the total hours of NLC during daylight (ionization) discarded due to missing MSE could be robust, and it would be an important result.

*Indeed, it would be very interesting to learn about missing MSE if the ice is confirmed. Unfortunately it is only a very limited data set. If cases with too low solar elevation are excluded, the large majority of NLC are accompanied by MSE.*

*From the total number of NLC profiles, about 35% are observed during solar elevation below 5°(i.e. presumably too low ionization), about 20% are too weak (ß< 0.3\*10^{-10} /m/sr, see Fig. 4 in Gerding et al., JGR, 2013). From the remaining part some fraction is discarded because radar is switched off. Then some few profiles remain, where either turbulence is missing, the ionization it still too low (unlikely), or the radar signal was below the threshold (lidar and radar signals depend different on particle size and number density). Given the complexity of MSE signal creation, we think only very limited conclusions can be drawn from the data set that is much smaller than the data set presented here.*

*We provide a more detailed description of the reasons to discard a large part of the NLC data (p. 7, l. 18-21).*

Representativeness for all NLC: I still wonder about the numbers, and the reason for the discrepancy between the NLC statistics of centroid height of 82.6 km and the MSE-selected NLC centroid height of 83.3 km. Looking at the distribution in Fig. 4a this seems to be a significant discrepancy not related to geophysical variance. The authors mention the removal of NLC during nighttime. This cannot be the reason, as Gerding et al. (2013) show rather slightly higher NLC centroid altitude during night, so this should not be an effect of diurnal variation. They mention the removal of weak NLC profiles during the beginning and end of the measurement, but this assumes they are at low altitude, which was not shown and which I doubt, as at polar latitudes NLC are brightest at low altitude. Then they mention the removal of very low NLC due to absence of MSE. Now this speaks for a strong coupling of NLC and MSE in line with the observations of coincident edges, and for a true (regarding physics) difference between NLC with and without MSE. Then the selection is not representative, but it does not have to be, this altitude-dependence of NLC regarding MSE conditions would rather be a major result of this study, if robust. Because Kaifler et al. (2011) found no such difference in NLC altitude due to PMSE selection, their table lists 82.1 km for both.

> *The difference between the mean peak height of the MSE-selected NLC (83.3 km, Fig. 4) and all 2010-2016 NLC (82.8 km, see p. 12, l. 17) is 0.5 km, i.e. half of the bin width used in the histograms. Both values are within the "most-probable" bin. We do not estimate this "significant", but see the NLC selected here as representative for all, within the geophysical variability and the context of this study. The diurnal variation of NLC altitudes described by Gerding et al. (2013) has indeed a minimum during the day. But the occurrence rate needs to be taken into account, which has a maximum in the early morning and a minimum in the evening. Indeed, bright NLC are typically lower than weak NLC. But this does not hold completely true when the NLC vanishes at low altitudes as can be seen in Fig. 2 a and b. The "removal of very low NLC due to absence of MSE" happens in a few cases. Unfortunately our data set is too small to conduct a systematic study.*

When the authors claim that 600 or 700 m is within the accuracy or variability or due to the limited size of the dataset, then this also applies to the upper edges, and it would be even more difficult to draw conclusions from these numbers given the large uncertainty. Then I would conclude that the dataset is yet too small to derive results that can aid understanding of the formation processes.

> *We see a geophysical variability of a few hundred meters, depending on selection of the data. We did an analysis of mean layer peaks, upper and lower edges for different sub-datasets with ~1000 NLC and MSE profiles (random sub-samples as well as sub-blocks). We found a distance of upper edges of NLC and MSE between 200 m and 700 m. Therefore we judge the difference to the situation at high latitudes significant, where the mean distance of upper edges is 3.3 km. We add a statement about the variability within subsets in the Discussion (p. 13, l. 11-15).*

The authors main conclusion is that mature NLC particles are advected to the observation site. While this may be true, and was claimed before, it is not totally clear if this conclusion is confirmed by the absence of MSE above the NLC layer (coincidence of upper edges) and if it can thus be strengthened by this study. For all we know, the particles could nucleate and grow within the NLC layer, or they could nucleate above the NLC layer but MSE could be suppressed by other mechanisms. So I am not yet convinced that the presented analysis allows for conclusive interpretation on this topic.

*Extensive nucleation within the NLC layer is unlikely because the supersaturation is typically too small. Typically, nucleation starts close to the mesopause, where supersaturation is largest. Ice clouds not visible as MSE may exist, but this would mean a systematic lack of turbulence in this height region (but not in the height of the NLC). We cannot exclude this due to lacking observations, but estimate it to be unlikely.*

I am looking forward to the authors thoughts and data on the points I mentioned. In case a second revision should be prepared, I give the following advice for improvements of the text. Especially in the conclusion a number of formulations are imprecise or incorrect:

p. 13, l. 20 and p. 15, l. 5: This should be expressed more carefully. There are no temperature measurements available at the lower edge at mid-latitudes. So the fact that the lower edges coincide hints at a large temperature gradient. This is not the same than expecting coincident edges on the assumption of a large temperature gradient.

*The reviewer is right that we do not have temperature measurements right below the NLC/MSE available for the events described here. But we have examined the temperature above and below the NLC earlier (Gerding et al., JGR, 2007) and found a saturation ratio of typically 0.1 about 1 km below the ice cloud. Even under climatological conditions with generally less pronounced gradients (Gerding et al., ACP, 2008) the temperature gradient at 80 km in summer results in a ~30% decrease of saturation within 200 m, assuming a constant water vapor pressure. We add "typically rising temperatures" to make clear that this assumption is well justified (p. 14, l. 12).*

p. 15, l. 8: "the layer is thinner": thickness was not evaluated, and no numbers were given for MSE-only. So we don't really know if it is thinner.

*We added a figure as suggested above. Additionally we changed "layer" to "ice cloud" to be more precise (p. 15, l. 18).*

p. 15, l. 10: "thin layer above" does this mean between the NLC top and the MSE top? But the smaller particles reach down to the NLC bottom. "no such layer at all" does that mean NLC without MSE, or coinciding upper edges?

*We rephrased to „Clouds that already exist long enough to form large particles (NLC) show only a thin layer of small particles (invisible for the lidar but visible as MSE) above the NLC at our site. Or they show no particles at all above the NLC, i.e. the upper edges of NLC and MSE coincide." (p. 15, l. 19 – p. 16, l. 3)*

p. 15, l. 12: "southward or northward or weak". There is no preference from the analysis, I'd say.

*We agree and changed the phrasing to "Meridional winds above the NLC do not show a preferential direction for the examined events." (p. 16, l. 4/5)*

p. 15, l. 13: if they don't grow to optically visible sizes, local formation of NLC is not possible

*We deleted "NLC".*

The language could be improved regarding expressions as "provides some rough information", "scattering happens only on structures" , "layers stretch much higher than", "revealed from PMSE observations", "particles have been grown to sizes", "we partly find very large agreement"

*We have rephrased these expressions*

p. 1, l. 19: For sure, PMSE have not been observed for several decades by human eye. I see that this happened during revision, but nevertheless strongly advise to pay attention to correctness of sentences.

*We have changed the phrasing (p. 1, l. 20-23).*

p. 2, l. 14: Thomas (2003) formulated a question, and Russell et al. (2014) make no reference to climate change at all. This topic might be too complex to be covered in half a sentence, so I suggest to remove this statement.

*We have removed the statement about climate change (p. 2, l. 15/16).*

p. 6, l. 3: the lower edges do not agree, they are plus or minus 1 km.

*We have removed the statement about the lower edges (p. 6, l. 3).*

p. 8, l. 9: indeed I can imagine this (Fig. 2d, 3:30 UT) to be a FOV effect

*We agree. Unfortunately a final answer on this can not be given.*

p. 8, l. 11: one could estimate the drift time given typical wind and FOVs, should be few minutes only.

*We added an example for the drift time under the assumption of 20 m/s wind speed (~3.5 min, p. 8, l. 14/15).*

p. 11, l. 20: my interpretation would be that there is no preference for any wind direction above the NLC layer.

*We agree that there is no significant preference, and changed the phrasing in the summary accordingly (p. 16, l. 5).*

Technical:

p. 1, l. 1: suggesting: "We combined ground-based lidar observations of noctilucent clouds (NLC) with colocated, simultaneous radar observations of mesospheric summer echoes (MSE) at a mid-latitude site in order to compare ice layer altitudes. While larger ice particles (> 10 nm) are directly observed by lidar, the echoes recorded by radar are created by a complex interplay of ice particles, ionization and turbulence. The combined lidar and radar dataset thus includes some information on the size distribution and history of the clouds. .."

> *We rephrased the first sentences to "We combined ground-based lidar observations of Noctilucent Clouds (NLC) with collocated, simultaneous radar observations of Mesospheric Summer Echoes (MSE) in order to compare ice cloud altitudes at a mid-latitude site (Kühlungsborn/Germany, 54° N, 12° E). Lidar observations are limited to larger particles (>10 nm), while radars are also sensitive to small particles (<10 nm), but require sufficient ionization and turbulence at the ice cloud altitudes. The combined lidar and radar data set thus includes some information on the size distribution within the cloud and by this on the 'history' of the cloud."*

p. 1, l. 3: "first comparative study": "first" could be removed, as it is not clear if the authors claim to do this for the first time, or if they plan a second study

*Changed, see above.*

p. 1, l. 6: "rough information"-> "Thus, some information on the size distribution and history of the cloud is included in the combined lidar and radar dataset." Just a suggestion

*Changed, see above.*

p. 1, l. 10: "We find no difference of the lower edges", as the accuracy of the radar is 300 m.

*New phrasing: "On average, there is no difference between the lower edges …".*

p. 1, l. 13: "stretch higher" -> reach higher?

*Changed.*

p. 1, l. 17: I would turn this around: "High-altitude MSE, usually indicating nucleation of ice particles, are rarely observed in conjunction with lidar observations of NLC at Kühlungsborn."

*We changed the phrasing. Thank you.*

p. 2, l. 6: "Later on" this might refer to the 1996 citation, but the 2017 citation is in between, so this is unclear.

*Changed to "Further studies revealed …". (p. 2, l. 6)*

p. 2, l. 9: "comprehensive interpretation" -> review of PMSE physics?

*Changed.*

p. 2, l. 9: "Though, NLC.." Thus, NLC and PMSE are both indicators for..

*Changed.*

p. 2, l. 10: "indirect information on temperature … atmosphere where other data is sparse."

*Changed.*

p. 3, l. 11: were grown, have grown?

*Changed.*

p. 3, l. 26: delete "daylight-capable"

*We prefer to keep this information, because daylight capability is essential for this study.*

p. 5, l. 25: "to allow for a radar backscatter signal"

*Phrasing without "for" was suggested by Reviewer 1 in first revision.*

[revised manuscript text omitted]